# Comprehensive exploration of the translocation, stability and substrate recognition requirements in VIM-2 lactamase

John Z Chen[1], Douglas M Fowler[2,3], Nobuhiko Tokuriki[1]*

[1]Michael Smith Laboratories, University of British Columbia, Vancouver, Canada; [2]Department of Genome Sciences, University of Washington, Seattle, United States; [3]Department of Bioengineering, University of Washington, Seattle, United States

**Abstract** Metallo-β-lactamases (MBLs) degrade a broad spectrum of β-lactam antibiotics, and are a major disseminating source for multidrug resistant bacteria. Despite many biochemical studies in diverse MBLs, molecular understanding of the roles of residues in the enzyme's stability and function, and especially substrate specificity, is lacking. Here, we employ deep mutational scanning (DMS) to generate comprehensive single amino acid variant data on a major clinical MBL, VIM-2, by measuring the effect of thousands of VIM-2 mutants on the degradation of three representative classes of β-lactams (ampicillin, cefotaxime, and meropenem) and at two different temperatures (25°C and 37°C). We revealed residues responsible for expression and translocation, and mutations that increase resistance and/or alter substrate specificity. The distribution of specificity-altering mutations unveiled distinct molecular recognition of the three substrates. Moreover, these function-altering mutations are frequently observed among naturally occurring variants, suggesting that the enzymes have continuously evolved to become more potent resistance genes.

**\*For correspondence:** tokuriki@msl.ubc.ca

**Competing interests:** The authors declare that no competing interests exist.

## Introduction

The rise of drug-resistant bacterial pathogens has been rapid and inevitable following the introduction of novel antibiotics to the clinic. Pathogens often acquired resistances through horizontal gene transfer (HGT) using mobile genetic elements carrying antibiotic resistance genes, such as plasmids or transposable elements (*Wright, 2019*; *Surette and Wright, 2017*; *Codjoe and Donkor, 2017*). Under constant selection pressure from antibiotic use, these resistance genes continuously evolve to improve their efficacy and alter and broaden their specificity to other classes of antibiotics (*Surette and Wright, 2017*; *Livermore, 2012*). Understanding the molecular mechanisms and evolutionary dynamics of antibiotic resistance genes is crucial to finding sustainable solutions against the future dissemination and evolution of antibiotic resistance, such as through predicting future evolution and aiding in antibiotic and inhibitor design (*Crofts et al., 2017*; *Tehrani and Martin, 2018*).

Metallo-β-lactamases (MBL), or class B β-lactamases, are one of the major sources for the spread of multi-drug resistance bacteria. MBLs are metal dependent hydrolytic enzymes that degrade a broad spectrum of the widely used β-lactam antibiotics, including 'last resort' antibiotics such as carbapenems (*Bebrone, 2007*). Plasmid borne MBLs, such as VIM, NDM, IMP, and SPM-types, have been particularly problematic as they can spread to different bacterial pathogens and have no clinically effective inhibitors (*Nordmann and Poirel, 2002*). All major MBLs have also been undergoing continual evolution; VIM-type MBLs have diversified up to 70 amino acid mutations (26% sequence difference) into over 50 isolated variants, and some variants seem to have developed new substrate

specificity (*Yan et al., 2001*; *Schneider et al., 2008*; *Bogaerts et al., 2012*; *Mojica et al., 2015*; *Martínez-García et al., 2018*). Much effort has been made to characterize the molecular mechanisms and identify key residues in several major MBLs using diverse biochemical and structural approaches (*Lauretti et al., 1999*; *Franceschini et al., 2000*; *Wommer et al., 2002*; *Jin et al., 2004*; *King and Strynadka, 2011*; *Baier and Tokuriki, 2014*; *Makena et al., 2016*; *González et al., 2016a*; *González et al., 2016b*; *Socha et al., 2019*; *Sun et al., 2018*). However, the contributions of the majority of residues in these enzymes remain unexplored, and little is known of the molecular mechanisms governing substrate recognition.

One way to resolve these questions is through comprehensive, large-scale characterizations of mutations affecting the MBLs' efficacy and specificity. Deep mutational scanning (DMS) is a recently developed method for the characterization of thousands of mutations within a protein using deep sequencing (*Fowler et al., 2010*; *Fowler et al., 2014*; *Rubin et al., 2017*). The resulting high-resolution and comprehensive mutational datasets provide invaluable information for deciphering a subset of mutations related to monogenetic disease (*Starita et al., 2015*; *Weile et al., 2017*; *Matreyek et al., 2018*), understanding evolutionary dynamics of proteins—including viral coat, antibiotic resistance genes and hormone receptors (*Starr et al., 2017*; *Stiffler et al., 2015*; *Steinberg and Ostermeier, 2016*; *Lee et al., 2018*)— as well as elucidating protein sequence-structure-function relationships (*Araya et al., 2012*; *Firnberg et al., 2014*; *Thyagarajan and Bloom, 2014*; *Klesmith et al., 2017*; *Roscoe et al., 2013*; *Romero et al., 2015*; *Gray et al., 2017*). In particular, conducting DMS on a protein of interest under varying conditions—for example against different substrates or in different environments—has further unveiled in-depth molecular details of a protein, such as residues contributing to substrate specificity (*Melnikov et al., 2014*; *Wrenbeck et al., 2017*) and protein–environment interactions (*Mavor et al., 2016*; *Noda-García et al., 2019*; *Thompson et al., 2019*).

In this work, we use DMS to characterize the functional behavior of all ~5600 single amino acid variants of VIM-2 against three classes of β-lactam antibiotics (ampicillin, cefotaxime, and meropenem) and at two different temperatures (25℃ and 37℃), and gain deep insights into the molecular and evolutionary determinants of VIM-2's behavior. We generate a series of comprehensive and high-quality datasets, and develop a global understanding of VIM-2 by identifying residues that are critical for its function, stability and/or substrate specificity. We also examine VIM-2's signal peptide—an often overlooked feature despite its importance in expression and transport. Moreover, we use the data to assess the resistance characteristics and rationalize evolutionary outcomes of the clinically isolated natural variants of VIM-type genes, revealing that several mutations in the natural variants are functionally beneficial and lead to changes in substrate specificity.

## Results and discussion

### Deep mutational scanning of VIM-2 metallo-β-lactamase

DMS was conducted on a library of VIM-2 variants, each encoding a single amino acid substitution. The wild-type (wt) VIM-2 (UniProt ID: A4GRB6) was sub-cloned into a custom pIDR2 vector with a chloramphenicol resistance marker, where VIM-2 expression is controlled by the constitutive AmpR promoter (*Supplementary file 1*); an extra Gly was inserted at position two to facilitate cloning (See 'Generation of a VIM-2 mutagenized library' in methods), which was also further mutated and selected. We constructed the library of all possible single amino acid variants of wtVIM-2 through PCR based saturation mutagenesis, where each codon position is mutated to an 'NNN' codon using restriction-free (RF) cloning (*Figure 1A*; *van den Ent and Löwe, 2006*); there are 5607 possible variants in the library ((20 a.a. + stop codons)×267 positions). The plasmids of mutagenized codon libraries were pooled into seven groups—six groups of 39 codons (117 bp, 819 variants in each group) and 1 group of 33 (99 bp, 693 variants)—so the mutagenized region of each group can be covered by Illumina NextSeq deep sequencing. We estimated the mutation rate of our library construction by determining the full sequence of 87 variants by Sanger sequencing: only one nucleotide substitution was found outside the intended codon—which corresponds to a mutation rate of $1.4 \times 10^{-5}$—while two other variants had a large insertion or deletion, which would be filtered out during the variant identification process (see 'Variant identification and noise filtering' in methods).

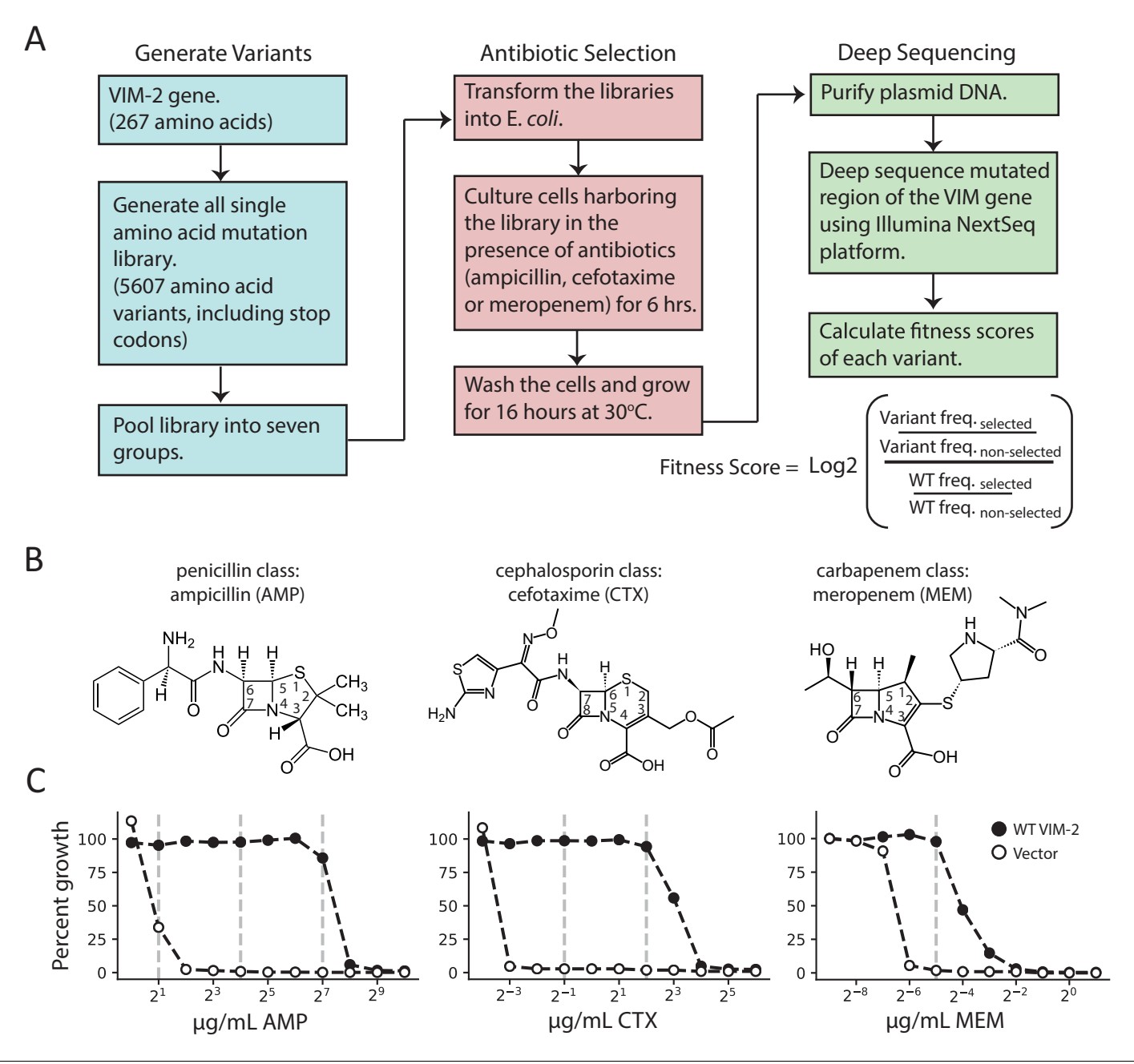

**Figure 1.** Deep mutational scanning (DMS) overview. (**A**) The workflow for DMS. All single amino acid variants are first generated using RF cloning, subsequently transformed into *E. coli* and then subject to selection for antibiotic resistance conferred to *E. coli*. The effects of selection (fitness score) were evaluated by deep sequencing and comparing the enrichment of each variant with and without selection. (**B**) Chemical structures of the antibiotics used in this study. (**C**) The dose-response growth curve of *E. coli* transformed with wtVIM-2 or an empty vector control for each antibiotic. Percent growth is calculated as $OD_{600}$ selected / $OD_{600}$ non-selected×100 after 6 hr of selection at 37°C. The vertical dashed lines indicate antibiotic concentrations at which selections were performed in this study. The dose-response curves of *E. coli* transformed with each of the seven library groups are in *Figure 1—figure supplement 1*. The number of variants represented in the mutated library is in *Table 2*. Other aspects of the cloning and data processing are described in *Figure 1—figure supplement 2* (PCR cloning method), *Figure 1—figure supplement 3* (deep sequencing error rates) and *Figure 1—figure supplement 4* (improving variant identification by an error filtering process).

The online version of this article includes the following figure supplement(s) for figure 1:

**Figure supplement 1.** Antibiotic dose-response curves of VIM-2 libraries for all antibiotics.

**Figure supplement 2.** Outline of steps for generating all single amino acid variants of VIM-2.

**Figure supplement 3.** Measurement of deep sequencing error rates using sequencing data of wtVIM-2 DNA.

*Figure 1 continued on next page*

*Figure 1 continued*

**Figure supplement 4.** Rationale and support for estimating deep sequencing noise for filtering variants observed in non-selected libraries.

Thus, we constructed a high quality variant library, comparable to other libraries constructed and deep sequenced in a similar manner (*Melnikov et al., 2014*).

Cultures of *E. coli* cells—specifically, *E. cloni* 10G, chosen for their high transformation efficiency and lack of *end*A and *rec*A—transformed with VIM-2 libraries (each group was treated separately) were subjected to antibiotic selection by incubating the culture at 37°C with LB media in the presence (selected) and the absence (non-selected) of three different classes of β-lactam antibiotics— ampicillin (AMP), a 3$^{rd}$ generation penicillin, cefotaxime (CTX), a 3$^{rd}$ generation cephalosporin and meropenem (MEM), a carbapenem (*Figure 1B*). To determine the selection conditions, the growth of *E. coli* cells harboring the plasmid encoding wtVIM-2 was examined at a range of antibiotic concentrations (1.0–1024 µg/mL AMP, 0.0625–64 µg/mL CTX, 0.002–2.0 µg/mL MEM) (*Figure 1C* and *Figure 1—figure supplement 1*). We chose to test the highest antibiotic concentration where wtVIM-2 can grow almost 100% relative to growth in media without β-lactam antibiotics, and at successive lower concentrations at 8-fold decrements where the range permits; selected conditions are 128, 16 and 2.0 µg/mL of AMP, 4.0 and 0.5 µg/mL CTX, and 0.031 µg/mL MEM (*Figure 1C*). The selection process for each antibiotic was conducted in duplicate on separate days. After selection, the plasmids were isolated, the mutagenized region of each group was amplified by PCR, and the amplicons were sequenced by the Illumina NextSeq 550 platform. The sequencing reads were error filtered, and the fitness score of each variant relative to wtVIM-2 was calculated using *Equation (1)*. (see methods for 'Deep sequencing and quality control').

$$fitness\ score = \text{Log2}\left(\frac{\frac{frequency\ of\ variant_{Selected}}{frequency\ of\ variant_{Non-selected}}}{\frac{frequency\ of\ wt_{Selected}}{frequency\ of\ wt_{Non-selected}}}\right) \tag{1}$$

Where the frequency of a variant (or wt) is the deep sequencing read count of the variant divided by the total reads in the corresponding sample. Variants with frequencies below the threshold of deep sequencing errors that was estimated from the deep sequencing of wtVIM2 (see methods for 'Variant identification and noise filtering') were excluded during scoring. The non-selected library shows excellent coverage, with 5535 of 5607 (98.7%) variants present after filtering in at least one replicate while 97.8% are observed in both replicates (*Table 2*). For selected libraries, we calculate the fitness score for any variants present in at least one non-selected replicate then average the fitness scores between the two selection replicates (see *Supplementary file 2A* for all fitness scores).

Our DMS experiments show high replicability in all conditions tested. The R$^2$ of a linear regression between variants observed in both replicates is 0.94 for selection with 128 µg/mL AMP, 0.91 for 4.0 µg/mL CTX and 0.85 for 0.031 µg/mL MEM (*Figure 2* and *Figure 2—figure supplement 1*). As expected, variants with synonymous mutations have near neutral fitness and variants with nonsense mutations (stop codons) have the lowest fitness scores. At the highest concentration of each antibiotic, variants with stop codons have fitness scores centered around −4 and lower, thus a fitness score of −4 is considered the lowest score cut-off for downstream analyses and fitness scores below this cut-off are set to −4 (*Figure 2*). Like previous DMS studies with other proteins, the overall fitness distribution of all variants exhibit a bi-modal distribution with a peak at neutral fitness and a long tail stretching toward negative fitness to another peak at the cutoff of −4 (*Figure 2*; *Stiffler et al., 2015*; *Firnberg et al., 2014*; *Roscoe et al., 2013*; *Jacquier et al., 2013*).

We confirmed the DMS fitness scores reflect the actual resistance level of variants (*Figure 2* and *Supplementary file 2B*). We isolated 45 unique variants (61 unique codons), and determined the half maximal effective concentration ($EC_{50}$) of *E. coli* culture harboring each variant for three antibiotics by measuring antibiotic dose-response curves. We fit the relationship using a sigmoidal function and identify a linear range of correlation for fitness scores within −3.1 to 0.1 for AMP ($EC_{50}$ 28–81 µg/mL), −2.7 to 0.6 for CTX ($EC_{50}$ 1.8–4.1 µg/mL) and −2.4 to 1.8 for MEM ($EC_{50}$ 0.012–0.066 µg/mL), which correspond to a 2.9, 2.3 and 5.5-fold range of $EC_{50}$ values for AMP, CTX and MEM, respectively (*Sebaugh and McCray, 2003*). All fitness scores outside the linear range are still qualitatively consistent with $EC_{50}$—where higher fitness scores correspond to higher $EC_{50}$ values and lower

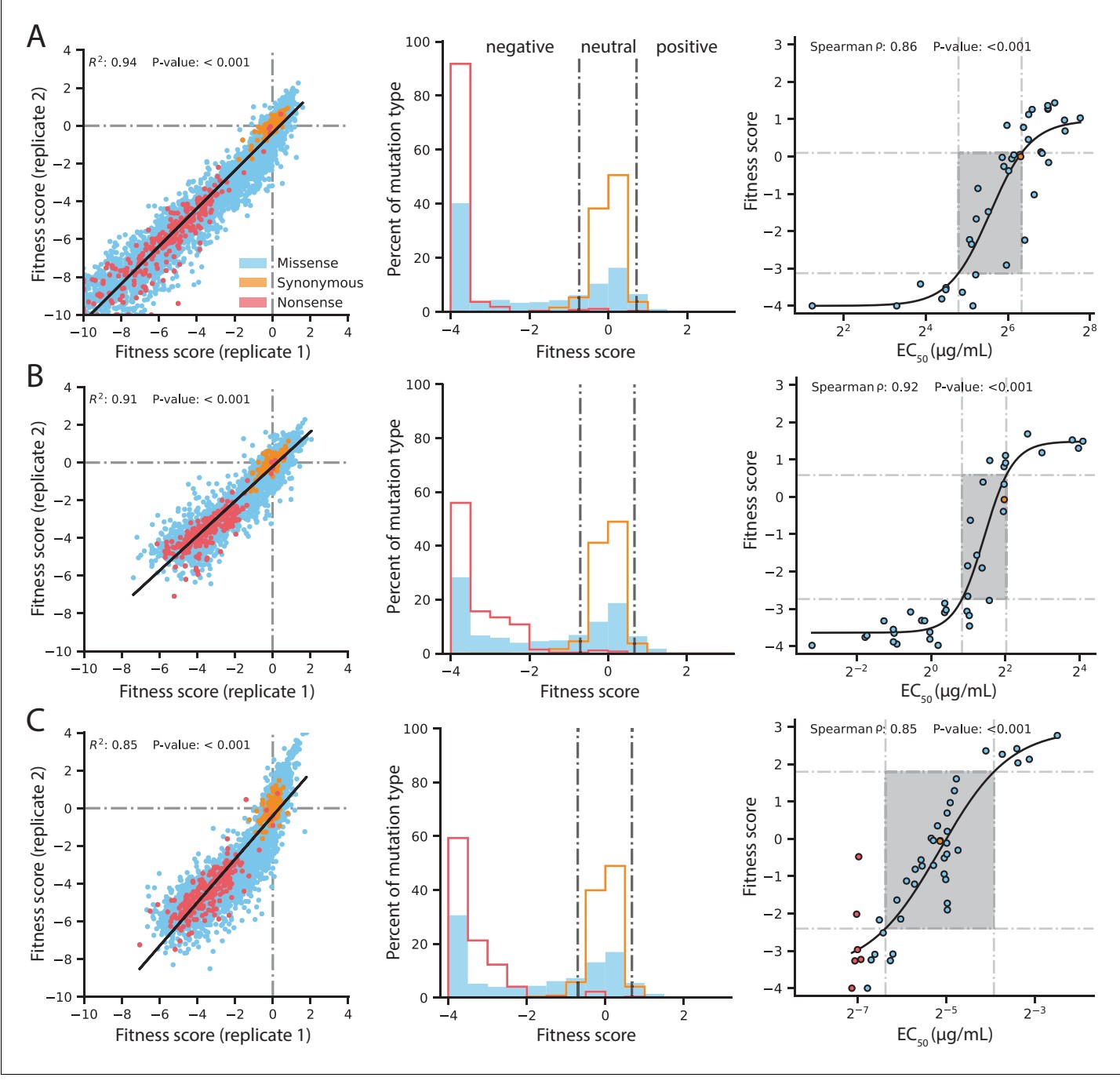

**Figure 2.** Quality control of DMS data and general mutational properties of VIM-2. In the horizontal panels, data are shown for (**A**) 128 µg/mL AMP selection, (**B**) 4.0 µg/mL CTX selection and (**C**) 0.031 µg/mL MEM selection. The color legend in panel **A**) is shared by all panels. For each horizontal panel, the left plot shows correlation between fitness scores of all variants in the two replicates of DMS; replicate correlation of selection conditions not shown here are in *Figure 2—figure supplement 1*. The middle plot shows distribution of fitness effects for all variants separated into synonymous, missense and nonsense distributions, where the vertical grey lines indicate fitness score cut-offs used to classify fitness effects as positive, neutral or negative. The proportion of variants in each fitness effect category can be found in *Figure 2—figure supplement 2*. The right plot shows the relationship of DMS fitness scores with antibiotic resistance ($EC_{50}$) of isolated variants measured in a dose-response curve; variants with resistance lower than the tested range could not be fitted for $EC_{50}$, leading to $EC_{50}$ values for 39 unique variants in AMP, 39 for CTX and 45 for MEM—some points are an average of the same codon or amino acid variant isolated multiple times. The filled rectangle in the background indicates the region of linear association between fitness scores and $EC_{50}$. The text at the top of each plot indicates the Spearman rank-order correlation coefficient and the P-value of the correlation. Individual $EC_{50}$ measurements can be found in *Supplementary file 2B*.

*Figure 2 continued on next page*

*Figure 2 continued*

The online version of this article includes the following figure supplement(s) for figure 2:

**Figure supplement 1.** Replicate correlation of fitness scores in the DMS experiments.

**Figure supplement 2.** Proportion of fitness effects for VIM-2 nonsynonymous variants selected in AMP, CTX and MEM.

fitness scores correspond to lower $EC_{50}$ values—which is supported by a Spearman rank-order correlation between fitness and $EC_{50}$ of at least 0.85 for selection using each antibiotic.

## Global view of VIM-2 enzyme characteristics

The fitness scores for variants selected at 128 μg/mL AMP are shown in *Figure 3* (See *Figure 3—figure supplements 1* and *2* for CTX and MEM); the inserted Gly2 is omitted to match the numbering for wtVIM-2 (For Gly2 fitness scores, see *Supplementary file 2A*). At a glance, there are several interesting trends in the DMS data of VIM-2. Variants with Cys mutations are highly deleterious throughout the catalytic domain (positions 27–266) (*Figure 3—figure supplement 3*). As wtVIM-2 possesses only one Cys for metal binding, additional Cys may cause the formation of undesired disulfide bonds, leading to misfolding (*Mehlhoff, 2020*). Pro variants are also often deleterious, as this residue disrupts secondary structures (*Stiffler et al., 2015*; *Firnberg et al., 2014*; *Gray et al., 2017*). We found 112 positions (42% of all residues) are highly sensitive to mutations, where over 75% of missense variants (excluding synonymous and nonsense mutations) display a fitness score $<-2.0$. These positions are likely key requirements for catalytic activity or protein stability and folding in wtVIM-2. Indeed, these positions include all six active-site metal coordinating residues (His114, His116, Asp118, His179, Cys198, His240), as well as 3–4 residues adjacent to each metal binding residue in the amino acid sequence that are likely to play important roles in the metal configuration and enzymatic function. Additionally, 92% of positions with high mutational sensitivity—including all metal binding residues—are located in the core of the protein (accessible surface area, ASA, of residue <30%) and 63% are almost completely buried (ASA <5%), congruent with previous findings (*Fowler et al., 2010*; *Stiffler et al., 2015*; *Thyagarajan and Bloom, 2014*; *Melnikov et al., 2014*; *Thompson et al., 2019*; *Kitzman et al., 2015*; *Figure 4A*). The association between mutational sensitivity and ASA is also evident at the level of individual variants, where the distribution of variants at positions with ASA <30% exhibits significantly lower fitness than the distribution of variants at positions with ASA ≥30% (two-tailed Mann-Whitney U test, P-value<0.001, *Figure 4A*).

To examine the distribution of fitness effects (DFE) of VIM-2 variants, we classified the 5291 nonsynonymous variants as having negative ($<-0.7$), neutral ($-0.7$ to $0.7$) or positive ($>0.7$) fitness by performing Z-tests of each variant's fitness scores against the fitness distribution of 244 synonymous variants (the null model distribution), adjusting for 5% false discovery rate using the Benjamini-Hochberg procedure. The DFE of VIM-2 variants is similar across all selection antibiotics at the highest screening concentration (128 μg/mL AMP, 4.0 μg/mL CTX, 0.31 μg/mL MEM), with ~65% of variants being negative, ~30% being neutral and ~5% being positive (*Figure 2—figure supplement 2*). The overall DFE also agrees with observations found in DMS of other enzymes, such as *E. coli* amidase, TEM-1 β-lactamase and levoglucosan kinase (*Stiffler et al., 2015*; *Firnberg et al., 2014*; *Klesmith et al., 2017*; *Wrenbeck et al., 2017*).

Next, in order to determine biophysical properties that explain the fitness scores, a linear model was constructed using the fitness score of variants as the response factor and various parameters such as ASA, ΔΔG predicted by Rosetta, change in amino acid volume and polarity, and the wt and variant amino acid states as predictors (*Table 1*, *Supplementary file 3A and 3B* for CTX and MEM models, *Supplementary file 2C* for data used in models). We first modeled each predictor alone with the response, then selected predictors that account for at least 10% of variance in the response ($R^2 >0.10$) and modeled them in combinations of 2 or more as individual terms and as interacting terms. The final model accounted for the greatest amount of variance using the fewest predictors—that is optimized for adjusted $R^2$—and combined four predictors (ASA, ΔΔG, wt and variant amino acid) capable of explaining 55% of the variation in fitness scores (adjusted $R^2 = 0.55$). The ASA alone can explain 21% of fitness score variation (*Table 1*) and ASA of the wt amino acid alone show a significant correlation ($R^2 = 0.50$) to the average fitness scores of the position, with mutations at exposed residues having less deleterious fitness effects on average (*Figure 4B*). The ΔΔG explains

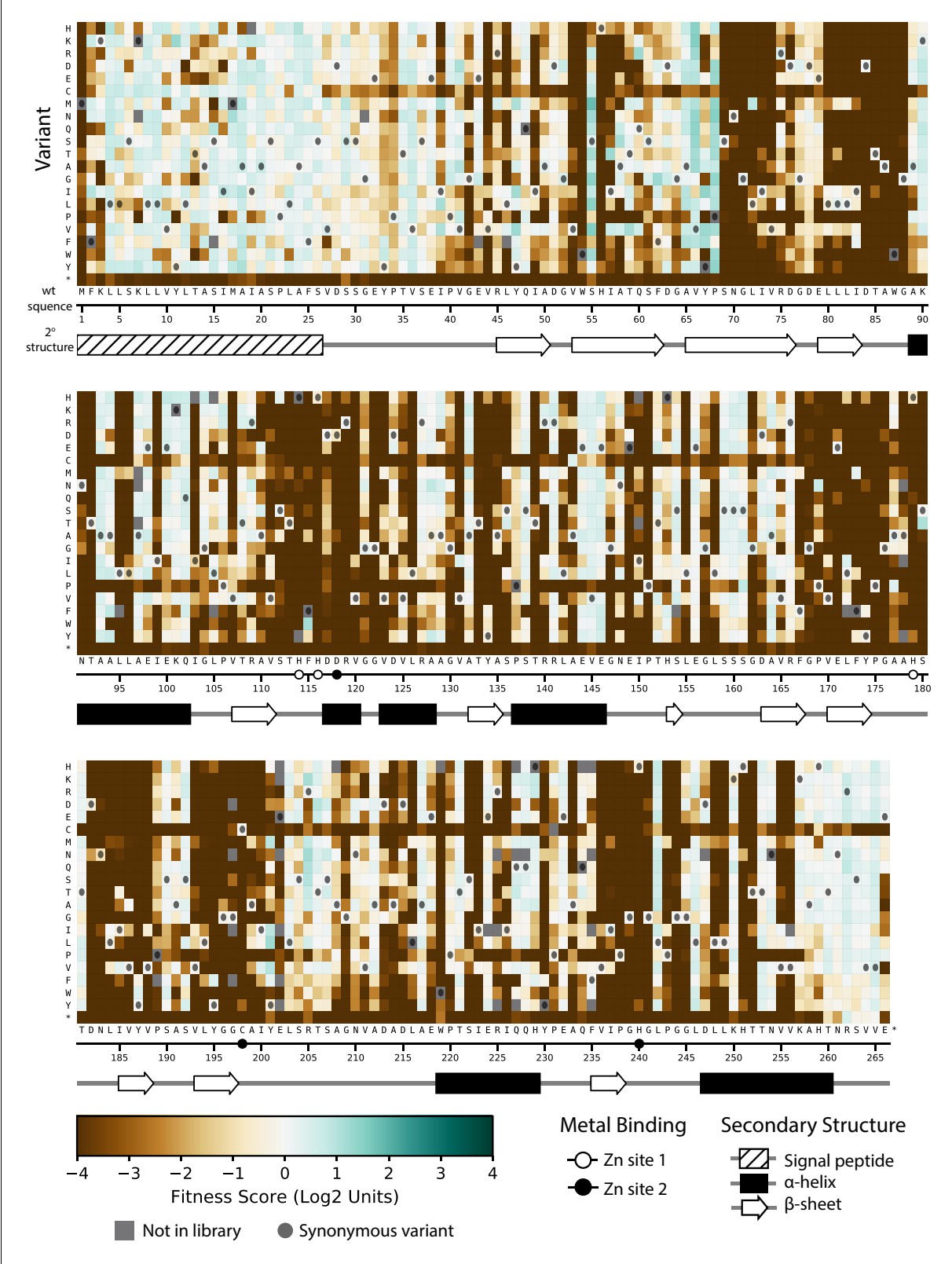

**Figure 3.** Fitness of all VIM-2 single amino acid variants under 128 µg/mL AMP selection. Each cell in the heat map represents the fitness score of a single amino acid variant. Synonymous variants are indicated by dark grey circles and variants that are not present in the library are in grey. The x-axis under the heat map indicates the wt residue and position (the six active site metal binding residues are highlighted as circles), while the y-axis indicates the variant residue at that position. The secondary structure of the wtVIM-2 crystal structure (PDB: 4bz3) is displayed below the heat map.

*Figure 3 continued on next page*

*Figure 3 continued*

Corresponding heat maps for variants under selection with CTX and MEM can be found in *Figure 3—figure supplement 1* and *Figure 3—figure supplement 2*, respectively. A comparison between the distributions of variants with each mutated amino acid can be found in *Figure 3—figure supplement 3*. All fitness data used in the heat maps can be found in *Supplementary file 2A*.

The online version of this article includes the following figure supplement(s) for figure 3:

**Figure supplement 1.** Fitness of all VIM-2 single amino acid variants under 4.0 µg/mL CTX selection.

**Figure supplement 2.** Fitness of all VIM-2 single amino acid variants under 0.031 µg/mL MEM selection.

**Figure supplement 3.** Comparison of fitness score by variant residue in 128 µg/mL AMP.

an additional 18% (*Table 1*) and there is overall correlation between ΔΔG and fitness score while individual predictions are relatively scattered, similar to previous findings that compared fitness to Rosetta folding energies or solubility scores (*Figure 4C*; *Firnberg et al., 2014*; *Klesmith et al., 2017*). Knowing the wt and variant amino acid can further explain another 10% and 5% of variation respectively (*Gray et al., 2017*; *Gray et al., 2018*). Thus, the results indicate structure and biophysical factors can explain the majority of fitness score tendencies.

## Codon and amino acid optimization in the signal peptide

The first 26 residues of VIM-2 has been identified as the signal peptide (*Lauretti et al., 1999*; *Franceschini et al., 2000*; *Garcia-Saez et al., 2008*), which is a sequence used to translocate the enzyme to the periplasm, then cleaved after transport (*Oliver, 1985*; *Pugsley, 1993*; *Freudl, 2018*; *Paetzel and Peptidases, 2019*). Our DMS data supports the known length of the signal peptide as mutations to Cys are much less deleterious before residue 26, suggesting these positions are not incorporated into the mature enzyme in the periplasm. In general, the signal peptide sequence has an amino terminal (N) region (residues 1–7) with one or more positive residues, a hydrophobic (H) region (residues 8–21) and a carboxy terminal (C) region (residues 22–26) that precedes the cleavage site containing a PXAXS motif (*Figure 5A*; *Oliver, 1985*; *Pugsley, 1993*; *Freudl, 2018*; *Paetzel and Peptidases, 2019*). The signal peptide is conserved at 17 of 25 positions (Met one excluded) across all VIM variants, and the remaining are binary differences between the conserved sequences of the VIM-1 and VIM-2 clades (*Figure 5A*, clades are defined by *Figure 8—figure supplement 1*).

Mutations in the signal peptide are generally tolerated (67% of missense mutations are neutral with 128 µg/mL of AMP) and even beneficial (10% of missense mutations are positive), which is consistent with a previous DMS study with TEM-1 (*Firnberg et al., 2014*; *Figure 5B*). Overall, the DFE for missense variants in the signal peptide is significantly more neutral than the DFE of missense variants in the catalytic domain (two-tailed Mann-Whitney U test, P-value<0.001, *Figure 5B*). In the N-terminal region, mutations to Lys3 are especially deleterious, likely due to the importance of a net positive charge in the N-region for efficient translocation (*Oliver, 1985*; *Inouye et al., 1982*; *Iino et al., 1987*). In contrast, Lys7 is tolerant to substitution—in fact, half of the natural VIM variants have a Ser at this position—suggesting that Lys3, rather than Lys7, is critical for translocation. In the H-region, residues Val10 through Ile16 are the most sensitive to mutation, especially when changed to a charged amino acid (*Firnberg et al., 2014*; *Mehlhoff, 2020*; *Oliver, 1985*). The C-region is most negatively affected by the mutation of Leu23 to Cys (fitness of −2.5) or Trp (−2.7), while 95% of variants are neutral or positive, including those in the PXAXS motif.

Interestingly, variants with evolutionarily conserved residues in the signal peptide often do not have the highest fitness, and we note both residue level and codon level dependencies in fitness (*Figure 5C*). Across the signal peptide, 10% of variants with missense mutations have positive fitness relative to wtVIM-2, with fitness ranging from 0.7 to 1.3. In terms of codon level fitness within synonymous variants, we find a small but statistically significant association between a mutated codon's change in mRNA folding energy (including the 5'UTR and excluding the 3'UTR) and fitness within mutants close to the start codon, supporting previous findings that avoidance of secondary structure near the start codon is favored (*Figure 5—figure supplement 3*, *Supplementary files 2D and 2E*, see Materials and methods for 'RNA folding energy calculation') (*Bhattacharyya et al., 2018*; *Bentele et al., 2013*). We also found that 73 of 143 'codon dependent variants'—variants in which any pair of synonymous codon scores differ by more than 2.0—are within the signal peptide, which

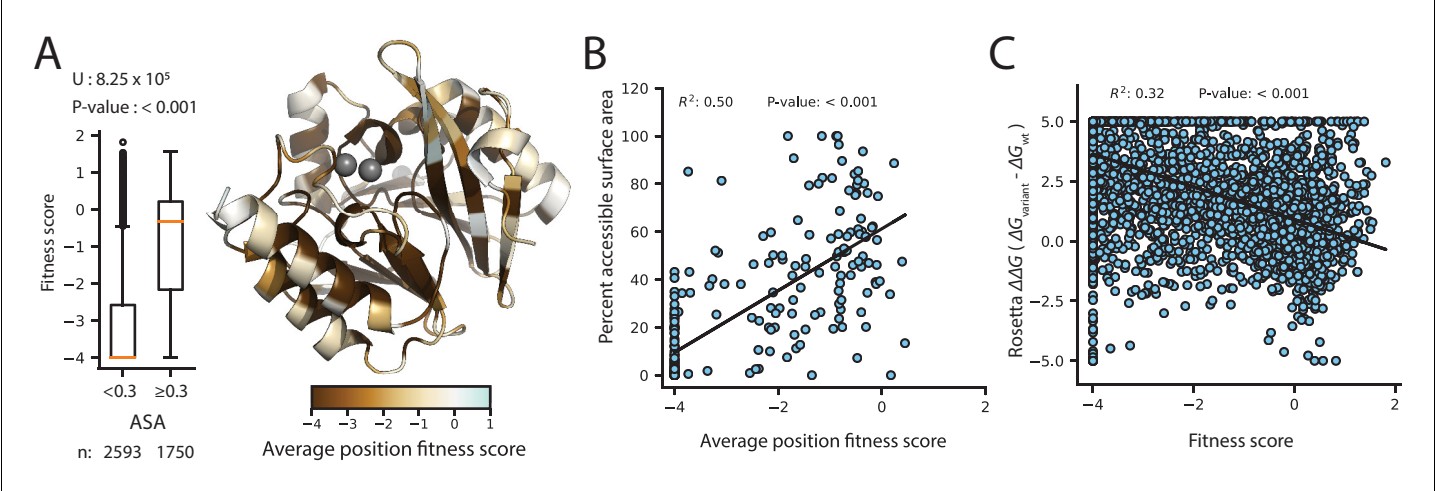

**Figure 4.** Correlation of fitness with structural attributes. Fitness scores are from DMS at 128 µg/mL AMP selection. (**A**) To the right, is the crystal structure of wtVIM-2 (PDB: 4bz3) colored by the average fitness of 20 amino acid mutations at each position. The inset to the left shows distributions of fitness scores for variants at positions with relative accessible surface area (ASA) <0.3 in wtVIM-2 or positions with ASA ≥0.3, with the number of variants in each distribution shown at the bottom and the results of a two-tailed Mann-Whitney U test between the two distributions at the top. (**B**) The correlation between accessible surface area and the average fitness of 20 amino acid mutations at each position. (**C**) The correlation between the changes in folding energy predicted by Rosetta and the DMS fitness scores for all variants.

is also suggestive of the importance of codon choice near the start of the coding region (*Kelsic et al., 2016*; *Figure 5—figure supplement 2*).

The less than optimal residue level fitness of wtVIM-2 may be because we employ *E. coli* as a host while natural VIM variants are often found in *Pseudomonas* (*Jia et al., 2017*), and/or the signal peptide is not selected to produce maximum expression in natural environments. It has been shown that different signal peptides produce variable expression levels and translocation rates for a given protein, both of which affect the final resistance, especially in different host organisms (*Socha et al., 2019*; *Mehlhoff, 2020*; *Inouye et al., 1982*; *Iino et al., 1987*; *Brockmeier et al., 2006*; *Mathiesen et al., 2009*; *Hemmerich et al., 2016*). Meanwhile, fitness variation at the codon level may be due to the presence of a different 5'UTR compared to the plasmid sequence found in clinical VIM-2 (*Bhattacharyya et al., 2018*; *Kelsic et al., 2016*). The inclusion of an extra Glycine at position two may affect the observed fitness, though this effect is expected to be small since signal peptides are highly variable in length and composition in the N-region (*Oliver, 1985*; *Pugsley, 1993*); 13/19 missense variants are neutral or positive at Gly2. Overall, the signal peptide is sensitive to changes in organismal and genetic context which would affect the outcome of horizontal gene transfer. Notably, the signal peptide is frequently mutated in naturally occurring VIM-type variants (see section on 'Natural VIM variation' below), suggesting changes in the signal peptide sequences may have played significant roles in dissemination of MBL genes to different hosts and adaptation to higher antibiotic concentrations.

## Elucidation of the role of residues in the catalytic domain

We sought to further examine the functional and structural roles of residues in the catalytic domain (positions 27–266) of wtVIM-2. We compare fitness scores of missense variants between selection in 128 µg/mL and 16 µg/mL AMP, as fitness at different AMP concentrations reflect a residue's degree of involvement in the enzyme's stability, expression and/or catalytic activity. Selection was also performed at 25°C in addition to 37°C to examine temperature dependent mutational effects, highlighting residues involved in protein folding and stability; in general, lower temperatures are permissive to variants with poor folding and high aggregation propensity while having a uniform effect on catalytic rate. To assess the role of each residue, we classified all positions in the catalytic domain into four types: *i*) 'tolerant', if 75% of variants are neutral even in the most stringent condition with 128 µg/mL AMP at 37°C, *ii*) 'essential', if 75% of variants are highly deleterious even in the least stringent condition with 16 µg/mL AMP at 25°C, *iii*) 'temperature dependent', if the difference in the fitness

**Table 1.** Linear model output for DMS fitness scores under 128 µg/mL AMP selection

| Predictor* | | Estimated effect† | Adjusted P-value (α = 0.05)‡ | Variance explained§ |
|---|---|---|---|---|
| (Intercept)¶ | | −1.73 | <0.001 | |
| ASA** | | 2.83 | <0.001 | 21% |
| Rosetta ΔΔG†† | | −0.28 | <0.001 | 18% |
| Starting(wt) residue | C | −1.53 | <0.001 | 10% |
| | D | −1.33 | <0.001 | |
| | E | −0.53 | <0.001 | |
| | G | −0.92 | <0.001 | |
| | H | −1.09 | <0.001 | |
| | I | −0.50 | <0.001 | |
| | L | −0.43 | <0.001 | |
| | N | −0.62 | <0.001 | |
| | R | −0.35 | 0.001 | |
| | S | −0.23 | 0.018 | |
| | V | −0.36 | <0.001 | |
| | W | −1.35 | <0.001 | |
| Variant residue | C | −1.69 | <0.001 | 5% |
| | D | −0.40 | 0.002 | |
| | P | −0.61 | <0.001 | |
| | W | −0.45 | <0.001 | |

*Each predictor indicates a class of wtVIM-2 derived values that were used as explanatory variables to model a linear relationship with the observed fitness score.

†The estimated effect is the predicted change in fitness score away from the intercept with a one unit increase in a continuous predictor or a binary change in a categorical predictor.

‡P-values indicates whether a predictor makes a significant contribution to the change fitness score, and are adjusted for a false discovery rate of 5% using the Benjamini-Hochberg procedure.

§The adjusted $R^2$ of each predictor when correlated with fitness, which is a measure of how much variation in the fitness score can be explained by each predictor in the linear model.

¶The intercept is the average fitness of all variants where the continuous variable is 0 (ASA and Rosetta ΔΔG) and the wt or variant residue is Ala.

**ASA ranges from 0.0 to 1.0.

††Rosetta ΔΔG ranges from −5.0 to 5.0 Rosetta energy units.

score between 25°C and 37°C is more than 2.0 in either 128 or 16 µg/mL AMP, and *iv*) 'residue dependent', if variants are temperature independent (similar scores at the two temperature) and exhibit a range of fitness rather than being mostly neutral or negative (*Figure 6A–B*, *Supplementary file 2F* for all classifications, also see 'Identification of critical residues and temperature dependence' in the Materials and methods).

As expected, the 55 'tolerant' positions are scattered around the surface of the protein and are mostly solvent exposed (80% of positions have greater than 30% ASA) (*Figure 6C–D*). The 20 'essential' residues include all six metal binding residues and deeply buried residues (70% have less than 5% ASA), which form the central core of the protein (*Figure 6C*). This core is further expanded into a larger scaffold by the 93 'temperature dependent' positions that are mostly buried in the structure (76% have less than 30% ASA) and largely hydrophobic—75% of the temperature dependent residues are non-polar (A, G, I, L, P, V) or aromatic residues (F, W, Y) compared to 58% for the entire catalytic domain. The 72 'residue dependent' positions tend to be near the surface or at packing interfaces between α-helices and β-sheets (*Figure 6D*).

The fitness of variants at temperature and residue dependent positions show equally strong association with the Rosetta predicted ΔΔG, indicating both classes of residues have contributions to structural packing (*Figure 6—figure supplement 1*). However, the location and hydrogen bonding

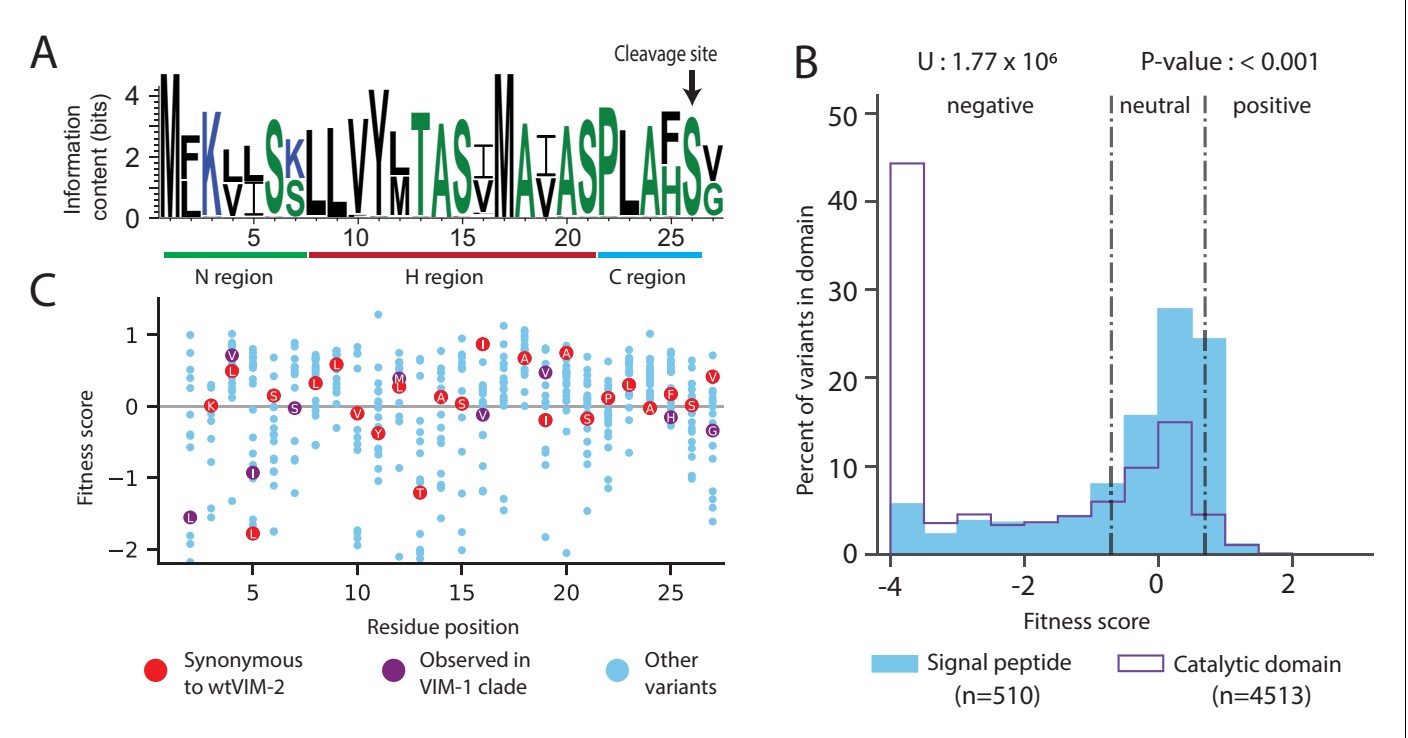

**Figure 5.** Conservation patterns and fitness scores in the signal peptide. (**A**) Sequence logo of the signal peptide region aligned across all VIM natural variants generated using WebLogo (https://weblogo.berkeley.edu/). Positions with two major naturally occurring residues are conserved differences between the VIM-1 and VIM-2 clades (clades are defined in *Figure 8—figure supplement 1*). (**B**) The distribution of fitness effects of all missense variants, separated into signal peptide variants and catalytic domain variants. The number of variants in each distribution are displayed in the legend below the distributions. The results of a two-tailed Mann-Whitney U test between the distributions are displayed above the distributions. (**C**) DMS fitness scores of all variants at each position of the signal peptide. Synonymous variants of wtVIM-2 and conserved variants observed in the VIM-1 clade are highlighted as labelled circles. Additional information on codon variant fitness in the signal peptide can be found in *Figure 5—figure supplement 1* (replicate correlation of codon variant fitness), *Figure 5—figure supplement 2* (heat map of codon variant fitness) and *Figure 5—figure supplement 3* (correlation between RNA folding energy and codon variant fitness).

The online version of this article includes the following figure supplement(s) for figure 5:

**Figure supplement 1.** Replicate correlation for fitness scores of all codon variants under selection in 128 μg/mL AMP.

**Figure supplement 2.** Heat map of codon variant fitness scores for the signal peptide of VIM-2 under selection in 128 μg/mL AMP.

**Figure supplement 3.** Correlation between the fitness score under selection in 128 μg/mL AMP and predicted ΔΔG of RNA folding for codon variants in the signal peptide.

behavior of each class of residues suggest different functional roles. Essential and temperature dependent residues display an enrichment of sidechain-backbone h-bonding relative to the proportion of h-bonding residues in each class (*Figure 6—figure supplement 2*, *Supplementary file 2G*), suggesting—when combined with the formation of a hydrophobic core—these residues are largely involved in protein folding and stability (*Roscoe et al., 2013*; *Thompson et al., 2019*; *Flynn, 2019*). In contrast, 'residue dependent' positions are prominent in the two major active site loops (14/23 positions from 60 to 68 and 201–214) and on packing interfaces on these loops' distal faces from the active site. The loop holding metal-binding residues His114, His116 and Asp118, and the helix positioning the loop into the active-site (positions 112–129) are also enriched in residue dependent positions (10/15 non-metal-binding positions), suggesting possible effects on metal and substrate positioning. Thus, many of the 'residue dependent' positions are likely to be involved in catalysis through direct or indirect substrate interactions, and also affect the overall shape of the active site.

## Distinct recognition for different classes of β-lactam substrates

VIM-2 is known for its broad spectrum activity against all classes of β-lactam antibiotics except monobactams, but how residues achieve substrate recognition remains unknown. We examined

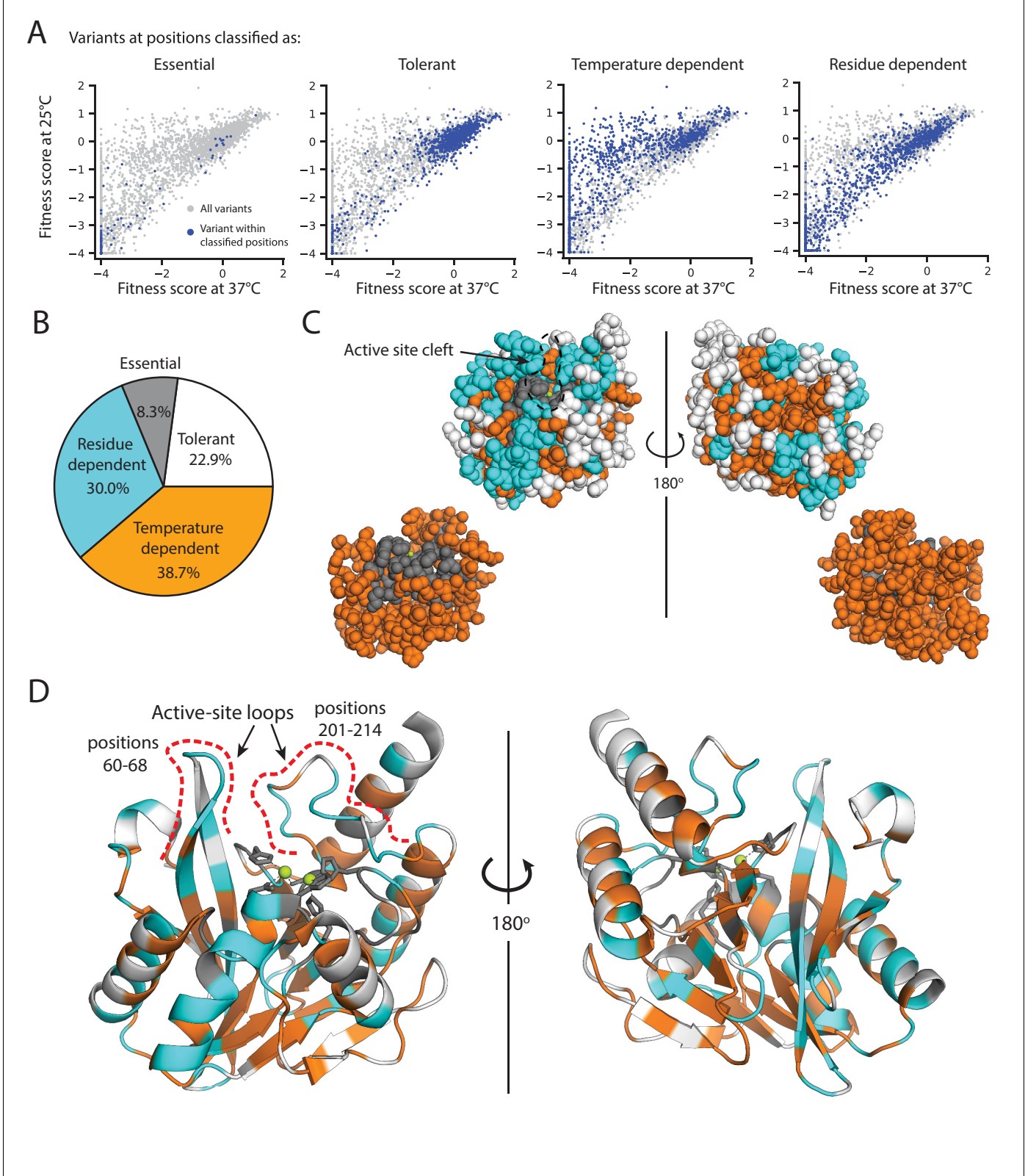

**Figure 6.** Distribution of mutational tolerance and temperature dependence of wtVIM-2 residues. (**A**) Scatterplots comparing fitness scores under selection at 25˚C and 37˚C (128 μg/mL AMP). Variants within the classified positions are highlighted in dark blue while all variants are plotted in grey for reference. (**B**) Proportion of residues in the wtVIM-2 catalytic domain that have been classified into each behavioral category. (**C–D**) The wtVIM-2 crystal structure (PDB: 5yd7) is colored by the behavioral classifications, the active-site zinc ions are colored in lime green. (**C**) View of the spacial distribution of

*Figure 6 continued on next page*

*Figure 6 continued*

temperature dependence in wtVIM-2, with residues depicted as spheres. The upper row depicts all residues, while the lower row depicts only essential and temperature dependent residues. (D) Cartoon representation of wtVIM-2 with metal-binding residues shown as sticks. All classifications can be found in *Supplementary file 2F*. Additional analysis of temperature dependence can be found in *Figure 6—figure supplement 1* (correlation of Rosetta ΔΔG with temperature dependence) and *Figure 6—figure supplement 2* (proportion of sidechain-backbone hydrogen bonding by temperature dependence).

The online version of this article includes the following figure supplement(s) for figure 6:

**Figure supplement 1.** Rosetta ΔΔ*G* in relation to temperature dependence classifications.

**Figure supplement 2.** Hydrogen bonding behavior in relation to temperature dependence classifications.

mutations that alter substrate specificity to identify wt residues responsible for substrate recognition by comparing fitness scores between the three antibiotics (128 µg/mL AMP, 4.0 µg/mL CTX, 0.031 µg/mL MEM). First, we identified 29 'globally positive' variants across 10 positions in the catalytic domain that increase fitness score to >1 in all antibiotics, which is more conservative than the cut-off of >0.7 for variants with positive fitness effects relative to wtVIM-2 and is above the upper fitness score range of the peak centered at neutral fitness in the DFEs (*Figure 2*; *Supplementary file 2H*). Residues at positions 47, 55, 66, 68 and 205 each give rise to at least three globally positive variants (24 total) while 57, 65, 115, 180 and 201 each give rise to one; 9/10 positions with globally positive variants are near the active site, having at least one atom within 15 Å of the active site zinc ions (*Figure 7A*). Next, we compare fitness scores of different antibiotics in pairs, and identified variants with a change in fitness effect classifications (negative, neutral or positive) combined with a 2.0 fitness score difference between antibiotics. We identified 78 specificity altering variants across 25 positions, with 23/25 positions near the active site (*Figure 7B*, *Table 3* and *Supplementary file 2I* for individual specificity variants, *Figure 7—figure supplement 1* for fitness heat maps at specificity positions). We confirm the specificity by comparing the fitness scores with the $\log_2(EC_{50\ var}/EC_{50\ wt})$ of the variant in the three antibiotics (*Figure 7—figure supplement 2*). Of the 25 positions, five are shared by both specificity and globally positive variants, and specificity changes are enhanced by the positive fitness at three of these positions (68, 201 and 205). However, most changes in specificity are due to decreases of fitness in one or two substrates, and only three variants (R205H/I/V) maintain neutral or higher fitness in all antibiotics (*Melnikov et al., 2014*; *Wrenbeck et al., 2017*). When examining the roles of these positions, we find nine 'residue dependent' and one 'tolerant' position, as expected for positions that interact with substrate rather than affect folding (*Dellus-Gur et al., 2013*). However, the other 15 positions are 'temperature dependent' positions, suggesting residues that are involved in substrate specificity are also embedded in the protein core.

**Table 2.** Variants observed in each library group.

| Group | Number of positions | Total possible variants | 37°C* | | 25°C* | |
| | | | Both replicates† | At least one replicate‡ | Both replicates | At least one replicate |
|---|---|---|---|---|---|---|
| 1 | 39 | 819 | 808 | 811 | 808 | 812 |
| 2 | 39 | 819 | 812 | 813 | 811 | 814 |
| 3 | 39 | 819 | 803 | 807 | 802 | 807 |
| 4 | 39 | 819 | 802 | 813 | 809 | 812 |
| 5 | 39 | 819 | 801 | 812 | 804 | 811 |
| 6 | 39 | 819 | 775 | 793 | 789 | 796 |
| 7 | 33 | 693 | 682 | 686 | 678 | 690 |
| Total | 267 | 5607 | 5483 | 5535 | 5501 | 5542 |
| % coverage | | | 97.8% | 98.7% | 98.1% | 98.8% |

*Non-selected libraries were grown, sequenced and filtered separately at 37°C and 25°C.

†The observed number of variants that passed noise filtering in both sequencing replicates of the non-selected library.

‡The observed number of variants that passed noise filtering in at least one sequencing replicate of the non-selected library.

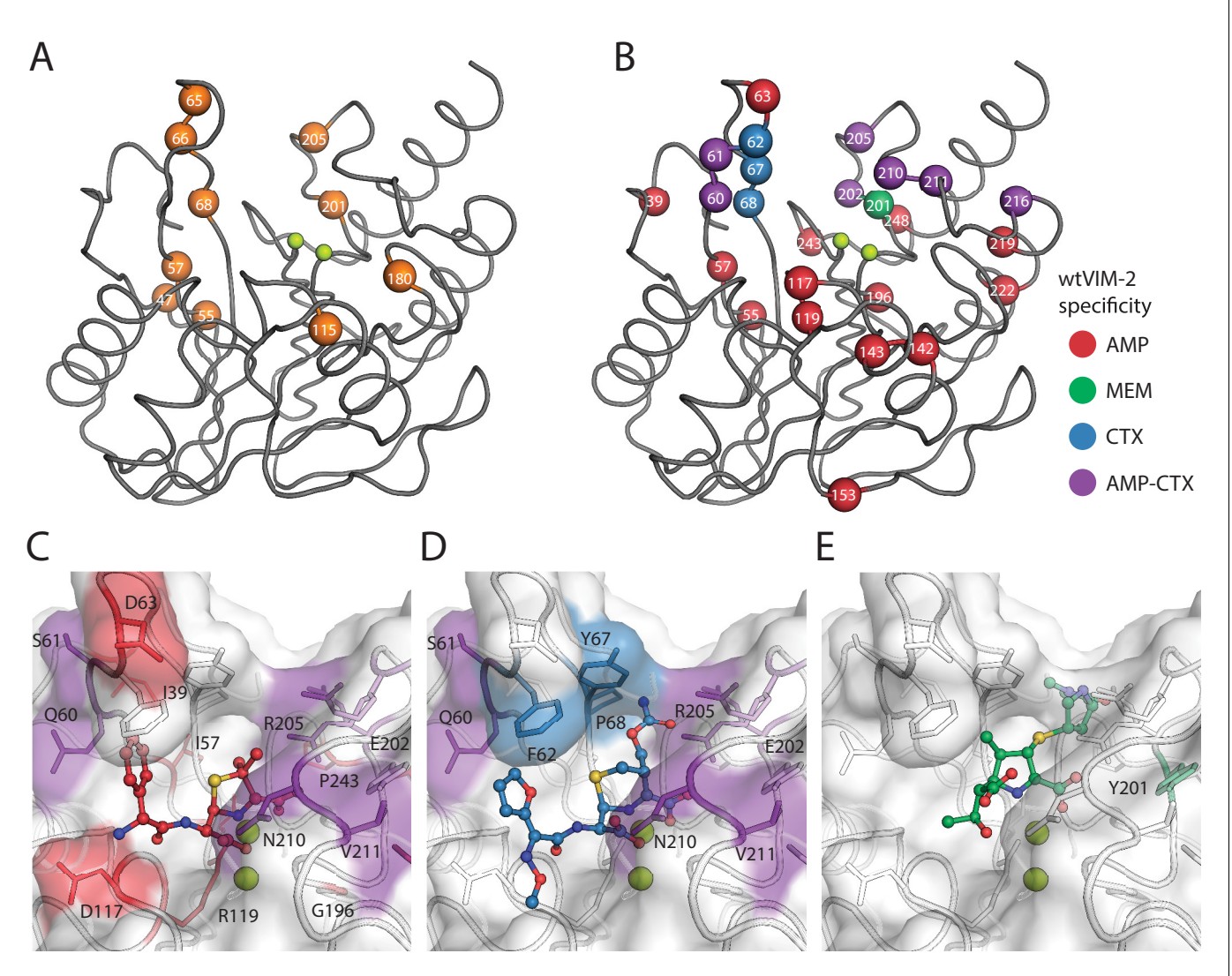

**Figure 7.** Visualization of VIM-2 specificity determining positions. The wtVIM-2 crystal structure (PDB: 5yd7) is featured in all panels, with the active site Zn ions depicted as lime green spheres. (A) Positions with at least one globally positive mutation are highlighted as orange spheres, which are also found in *Supplementary file 2H*. (B) Positions classified as being responsible for specificity towards certain antibiotics in wtVIM-2 are color coded by antibiotic and highlighted as spheres; the positions are listed by specificity in *Table 3*. Individual variants classified as having changes in specificity are listed in *Supplementary file 2I*. Heat maps of specificity positions for fitness under selection in each antibiotic and fitness differences between antibiotics can be found in *Figure 7—figure supplement 1*. (C–E). Close-up views of the specificity residues in the active site with (C) hydrolyzed ampicillin (PDB: 4hl2), (D) cefuroxime (PDB: 4rl0) and (E) meropenem (PDB: 5n5i) from VIM-1 and NDM-1 structures that have been aligned to the wtVIM-2 structure using the active site residues. Residues are colored by the inferred substrate specificity as in B). Substrates are shown in stick and ball representation. Comparison of fitness changes with $EC_{50}$ changes in individual variants are found in *Figure 7—figure supplement 2*.

The online version of this article includes the following figure supplement(s) for figure 7:

**Figure supplement 1.** Fitness landscapes of VIM-2 at substrate specificity positions.
**Figure supplement 2.** Comparison of fitness scores and $EC_{50}$ of specificity variants.

Interestingly, there is a strong bias in specificity changes depending on mutations and their positions in the active site. In 21 of 25 positions, specificity variants decrease AMP fitness, while in 10 positions variants decrease CTX fitness and variants decrease MEM fitness at only one position (*Figure 7B*). This bias is maintained at the level of individual specificity variants, where 96% decrease the fitness in either AMP and/or CTX—29 only decrease AMP, 33 only decrease CTX and 10 decrease both—while only three variants decrease MEM. The residues specific to AMP—where

mutations to the residue decrease AMP fitness, but are neutral for CTX and/or MEM—are spread around the active site, including the two active site loops (60–68 and 201–214) as well as residues in the protein scaffold. In contrast, the residues specific for CTX are restricted to the two active site loops (*Mojica et al., 2015*; *Martínez-García et al., 2018*; *Moali et al., 2003*; *Merino et al., 2010*; *Leiros et al., 2014*; *Leiros et al., 2015*); mutations in positions 62, 67 and 68 are fully specific to CTX, while positions 202, 205, 210 and 211 all have mixed specificity for CTX and AMP. To visualize residue-substrate interactions, we overlaid AMP, cefuroxime and MEM substrates in the active site of VIM-2 through alignment with VIM-1 and NDM-1 structures crystallized in complex with these substrates. Some substrate interactions are apparent from proximity, such as the packing of hydrophobic residues in loop 60–68 to the non-polar, aromatic substituents on the AMP (C6) and cefuroxime (C7) that is missing in most carbapenems (*Figure 7C–D*). However, Glu202 and Arg205 seem to be in better position to interact with the C2 substituent of MEM and are further from AMP or cefuroxime, yet both residues are specific for AMP and CTX. Moreover, many residues in the protein scaffold that are affecting AMP specificity do not directly interact with the substrates. We hypothesize that these distant residues may contribute to solvent related phenomenon—such as displacement of solvent and/or bridging solvent with ligand to affect substrate binding energy (*Maurer and Oostenbrink, 2019*; *Spyrakis et al., 2017*)—or alter protein dynamics to affect substrate specificity (*González et al., 2016b*; *Petrović et al., 2018*; *Campbell et al., 2018*; *Campbell et al., 2016*; *Singh et al., 2015*).

Although wtVIM-2 degrades all three β-lactams, our observations suggest that the enzyme interacts with each substrate in a different manner. AMP interacts with many residues around the active site and is the most sensitive to mutations, while CTX specificity relies exclusively on interactions with residues in the active site loops. Interestingly, MEM seems to rely on contacts shared with other antibiotics, which suggest that carbapenem resistance of VIM variants may have coevolved with other antibiotics.

## Natural VIM variation favors neutral, adaptive and specificity mutations

Currently, 56 unique VIM-type MBL sequences (including wtVIM-2) have been found on plasmids in β-lactam resistant clinical isolates (*Supplementary file 2J*; *Martínez-García et al., 2018*; *Jia et al., 2017*). The DMS data of VIM-2 enable us to characterize these naturally occurring mutations comprehensively. We classified these sequences into four clades, represented by VIM-1 (between 25–29 mutations from VIM-2 each, 45 unique mutations total), VIM-2 (1–6 mutations, 31 total), VIM-7 (70 mutations), and VIM-13 (32–33 mutations, 34 total) (*Figure 8A*, *Figure 8—figure supplement 1*), with 131 unique point mutations relative to VIM-2 across 99 positions (*Figure 8—figure supplement 2*, *Supplementary file 2K*).

As expected, the fitness distribution of naturally occurring mutations in the VIM-type MBL for all antibiotics (128 µg/mL AMP, 4.0 µg/mL CTX, 0.031 µg/mL MEM) shows enrichment for neutral and positive mutations (*Figure 8B*). The trend suggests that natural variants have adapted to higher resistance as 31% of all 'globally positive' catalytic domain variants in DMS are also naturally occurring mutations, and 17% of all natural mutations have positive fitness effects for at least one antibiotic. At least 66% of variants have neutral fitness effects within each antibiotic. Natural mutations are disfavored in residues crucial for activity or stability (*Figure 8C*): Of the 99 mutated positions, only a small proportion of 'essential' (1/20 positions) and 'temperature dependent' (21/93) positions have been mutated while large proportions of the signal peptide (20/26) and 'tolerant' positions (32/55)

**Table 3.** Inferred specificity of residues in wtVIM-2.

| Specificity | Positions* |
|---|---|
| AMP | **39**, **55**, **57**, 63, 117, **119**, **142**, **143**, **153**, **196**, **219**, **222**, **243**, **248** |
| CTX | 62, 67, <u>68</u> |
| AMP or CTX | **60**, 61, 202, **205**, 210, 211, 216 |
| MEM | **201** |

*Positions in bold are temperature dependent. The underlined position is tolerant. The unformatted positions are residue dependent.

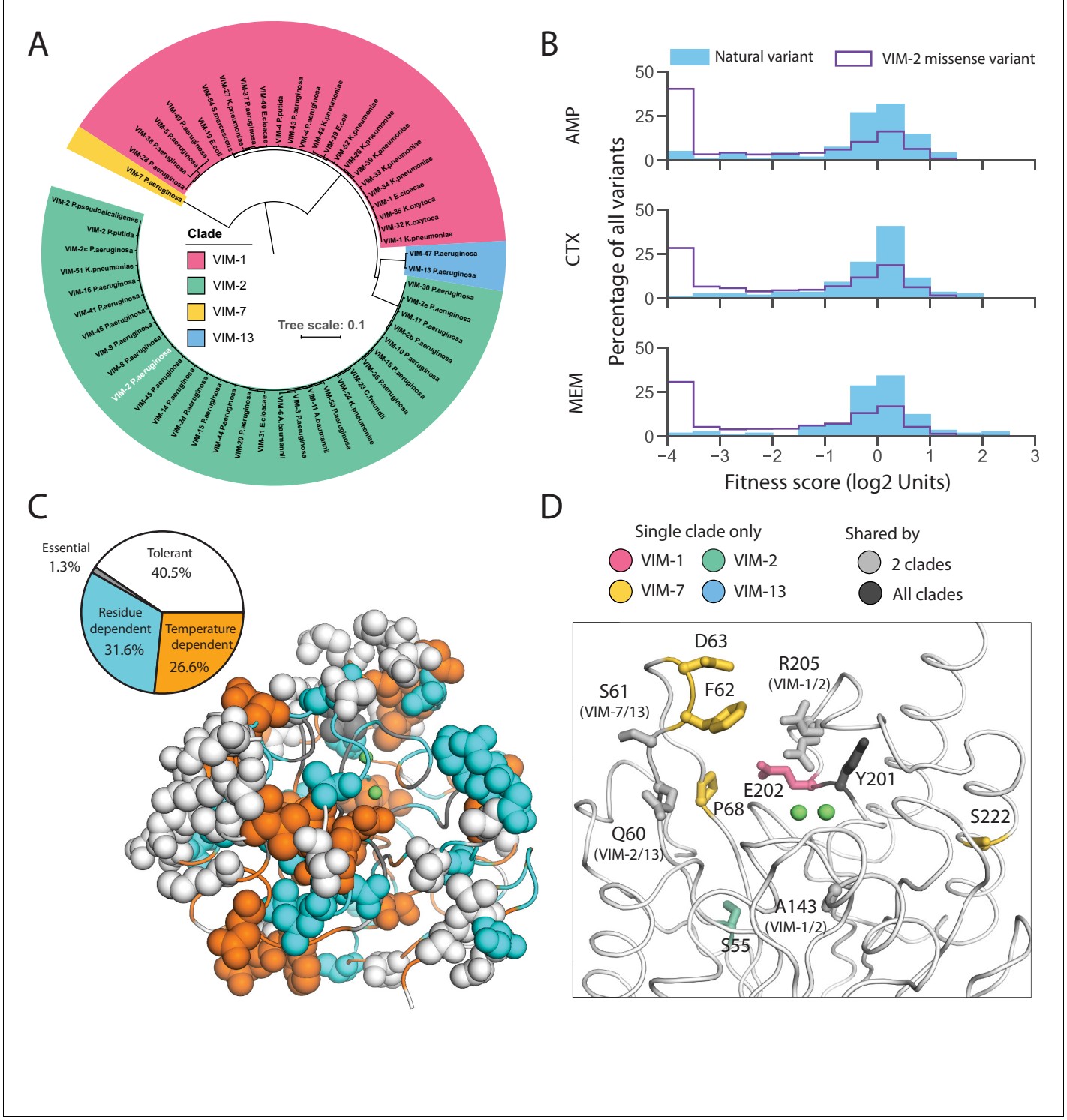

**Figure 8.** Behavior of natural VIM variants inferred from DMS fitness. (A) Maximum likelihood phylogenetic tree of all natural VIM variants examined in this study, colored by major clades (a larger version of the tree is presented in *Figure 8—figure supplement 1*). The wtVIM-2 sequence is highlighted in white. (B) Distribution of fitness for all unique individual mutations found in VIM natural variants compared to all missense variants measured in DMS for all three antibiotics. (C) All residues mutated in the natural variants are shown in sphere representation and colored by mutational tolerance and temperature dependence. The pie chart on the left shows the proportion of natural variant positions in each classification. (D) wtVIM-2 residues that are both mutated in at least one natural variant and affect specificity are highlighted as sticks, colored by the clade(s) in which the residue is mutated. All positions mutated in natural VIM-type variants relative to wtVIM-2 can be found in *Figure 8—figure supplement 2*.

The online version of this article includes the following figure supplement(s) for figure 8:

*Figure 8 continued on next page*

*Figure 8 continued*

**Figure supplement 1.** Maximum likelihood phylogenetic tree of wtVIM-2 and 55 VIM-type variants.
**Figure supplement 2.** Mutated positions in natural VIM variants.

have been mutated. Interestingly, 44% (11/25) of specificity altering positions have been mutated, which suggests that VIM variants may have changed their substrate specificity during evolution (*Figure 8D*).

However, 10% of mutations are still highly deleterious (fitness score $<-2.0$) in 128 μg/mL AMP (6.9% for 4.0 μg/mL CTX, 6.1% for 0.031 μg/mL MEM), indicating other factors that affect natural variation (*Figure 8B*). These 13 deleterious mutations are spread over 11 positions, where one position is in the signal peptide, and 10 are in the catalytic domain. The signal peptide mutation (K3Q) occurs only in VIM-7 and eliminates the Lys3 critical to translocation, but this is likely neutral as VIM-7 has a S6R mutation that replaces the positive charge. Two mutations are only deleterious to AMP (Q60H, A143T), thus altering substrate specificity. Furthermore, we suspect the neutral I185V mutation (all 55 natural variants have Val185 while our wtVIM-2 has Ile185) acts as a global suppressor (*Brown et al., 2010*; *Huang and Palzkill, 1997*), and permit the accumulation of four mutations that are deleterious in all antibiotics (T139A, T139I, V236G and V255A) as I185V is the only other mutation in these natural variants. The remaining six deleterious mutations are likely neutralized by specific intramolecular epistasis, or the background dependence of mutational effects (*Starr and Thornton, 2016*; *Miton and Tokuriki, 2016*; *Breen et al., 2012*; *Sarkisyan et al., 2016*; *Pokusaeva et al., 2019*), as these mutations occur in natural variants with at least 25 mutations relative to VIM-2. While such epistasis will hamper our ability to perfectly predict the effect of mutations, the VIM-2 dataset presented in this study contributes to further our understanding of MBL evolution, and can help orient our predictions concerning the emergence of future resistance.

## Conclusion

In this work, we report the first comprehensive mutational analysis of an enzyme in the MBL, class B β-lactamase family, one of the most important enzyme families underlying the dissemination of multi-drug resistance to pathogens. We uncover a sensitivity to variation in the signal peptide of VIM-2, that may be due to codon dependent RNA folding or incompatibility with the host translocation system. Such findings highlight the importance of genome and host context in resistance gene compatibility (*Socha et al., 2019*; *Bentele et al., 2013*; *Hemmerich et al., 2016*). By performing DMS at various conditions, three different antibiotics, and two temperatures, we enhance the understanding of sequence-structure-function relationships by unveiling a set of mutations for protein stability, catalysis and substrate specificity of VIM-2.

We find VIM-2's substrate specificity altering residues to be enriched near the active site, which enables us to elucidate the molecular basis of enzyme-substrate interactions. This finding is in contrast to a previous DMS study that tested multiple substrates: specificity-altering mutations of *E. coli* amidase (amiE) were distributed across the entire structure in a global mode of specificity determination (*Wrenbeck et al., 2017*). Thus, it is likely that each enzyme takes different mechanisms for recognizing diverse substrates. The monomeric MBLs have large, solvent-exposed active site clefts to recognize a wide range of substrates, while amiE which has a small, occluded active site and a hexameric quaternary structure that potentially favors controlling specificity through packing and subunit interactions (*Wrenbeck et al., 2017*). However, determinants of reaction enantioselectivity in 4-OT—another enzyme active as a hexamer—are concentrated in the active site, which further highlights unique behaviors in different enzymes (*van der Meer et al., 2016*). Regardless, understanding distinct mode of enzyme-substrate interactions will lead to design and development of new antibiotics and inhibitors to re-sensitize the MBL enzymes.

We have demonstrated that VIM-type variants have been continuously evolving by enhancing their resistance as well as altering their substrate specificity in nature. The study of natural variation also reinforces the observation that mutations found to be neutral or beneficial in an experimental setting tend to be enriched in nature as well (*Lee et al., 2018*; *Flynn, 2019*). The tendency for mutations at specificity positions to cause collateral sensitivity to AMP and CTX hints at some possibility for a combined treatment using carbapenems as the main antibiotic with penicillins or

cephalosporins added as supplement. Though the possibility of a combined treatment will require extensive investigation using specific β-lactams used in the clinics, and for tests to be conducted on clinically isolated strains that often contain multiple sources of β-lactam resistance including other MBLs like NDM-1 and serine β-lactamases like KPC-1 (*Monogue et al., 2018*). Given that we also observe natural variants with globally positive mutations, it is doubtful that use of β-lactams alone will be a long-term solution and inclusion of antibiotics with different mechanisms and β-lactamase inhibitors will be required to sustain effective treatments, combined with general surveillance and mitigation of spread for resistant pathogens (*Codjoe and Donkor, 2017*; *Crofts et al., 2017*). It is likely that many new VIM variants will emerge in the future, and our results will provide a valuable basis to predict likely mutations and estimate the resistance of newly found variants.

# Materials and methods

## Key resources table

| Reagent type (species) or resource | Designation | Source or reference | Identifiers | Additional information |
|---|---|---|---|---|
| Gene (*Pseudomonas aeruginosa*) | wtVIM-2 | UniProt | UniProtKB:A4GRB6 | |
| Strain, strain background (*Escherichia coli*) | E. cloni 10G | Lucigen | 60061 | Electrocompetent cells |
| Recombinant DNA reagent | pIDR2-wtVIM-2 | this paper | | Plasmid housing the wtVIM-2 sequence, maintained by the Tokuriki lab. See *Supplementary file 1* for full sequence. |
| Sequence-based reagent | primers with Nextera transposase adapter sequence | this paper | | Primers used to extract regions of the VIM-2 gene from variant libraries after a selection experiment, attaching the Nextera transposase adaptor sequence in the process. See *Supplementary file 2L* for all primer sequences. |
| Chemical compound, drug | Ampicillin | Fisher Scientific | BP1760 | |
| Chemical compound, drug | Cefotaxime | Fisher Scientific | BP29511 | |
| Chemical compound, drug | Meropenem | Sigma Aldrich | M2574 | |
| Commercial assay or kit | NextSeq 500/550 High Output Kit (300 cycles) | Illumina | 20024908 | |
| Software, algorithm | DMS-FastQ-Processing script | this paper | | Script used to merge and quality filter paired end FastQ reads. Code available at https://github.com/johnchen93/DMS-FastQ-processing (*Chen, 2020*; copy archived at https://github.com/elifesciences-publications/DMS-FastQ-processing) |

## Materials

LB Broth, Miller (BP1426), ampicillin sodium salt (BP1760) and cefotaxime sodium salt (BP29511) were purchased from Fisher Scientific. Meropenem trihydrate (M2574) was purchased from Sigma-Aldrich (Millipore sigma). E. cloni 10G electrocompetent cells (60061) and E. cloni 10G chemically competent cells (60107) were purchased from Lucigen Corp. The KAPA HiFi PCR Kit (KK2102) was purchased from KAPA Biosystems Inc, the E.Z.N.A. Cycle Pure Kit was purchased from OMEGA Bio-

tek Inc and the QIAprep Spin Miniprep Kit was purchased from Qiagen. The NextSeq 500/550 High Output Kit (300 cycles) (20024908) was purchased from Illumina Inc.

## Generation of a VIM-2 mutagenized library with all possible single amino acid substitutions

The wild-type (wt) VIM-2 gene including its signal peptide sequence from *Pseudomonas aeruginosa* was synthesized (Bio Basic Inc) and subcloned into an in-house plasmid, pIDR2 (chloramphenicol resistance) (*Supplementary file 1*), under a constitutive AmpR promoter using *Nco* I and *Xho* I restriction enzymes (Fisher Scientific). The ATG codon in the *Nco* I site was used as the start codon. However, the cut site requires an extra G nucleotide to follow the start codon and an additional Gly (GGA codon) residue was inserted into the second position of the VIM-2 sequence; this extra Gly relative to wtVIM-2 will be labeled as G2 to distinguish it from position two in the wt sequence. The pIDR2 plasmid containing the wtVIM-2 gene will be referred to as pIDR2-wtVIM-2.

To generate all single amino acid variants, a library of codon mutants was made for each codon (267 positions) in the wtVIM-2 gene using restriction-free cloning (RF cloning) (*van den Ent and Löwe, 2006*; *Figure 1—figure supplement 2*). We designed a forward primer for each codon position that contains a degenerate 'NNN' codon—using a MATLAB script used to design primers for the PFunkel method (*Firnberg and Ostermeier, 2012*)—and a single reverse primer, allowing a PCR to amplify part of the wtVIM-2 gene while incorporating the mutation. The PCR reaction to amplify part of the gene was done using a KAPA HiFi PCR Kit (Kapa Biosystems, Inc) for 30 cycles of amplification each with denaturation at 98°C for 20 s, annealing at 62°C for 15 s and extension at 72°C for 15 s; 1 ng of pIDR2-wtVIM-2 was used as template in a 20 µL reaction, with 1 µL each of the forward and reverse primer (10 µM). The first PCR products were purified using E.Z.N.A. Cycle Pure Kit (OMEGA Bio-tek, Inc). Afterwards, 10 µL of the first PCR product was then used as a primer to extend the entire plasmid, where the cycling conditions were identical to the first reaction except the extension time (90 s) and 1 ng of pIDR2-wtVIM-2 was freshly added as the template. Product from the second PCR was treated with *Dpn* I for one hour at 37°C to degrade the original wtVIM-2 plasmids, and then the amplified plasmids were purified and concentrated by the ethanol precipitation method. Subsequently, the purified plasmids were transformed into E. cloni 10G chemically competent cells (Lucigen Corp.) using the supplier's recommended heat-shock transformation protocol and plated on LB-Cm (containing 25 µg/mL chloramphenicol) agar plates. We then counted the number of colony forming units (CFU) obtained after the transformation for every mutagenesis library. Using CASTER (*Reetz and Carballeira, 2007*) and GLUE (*Firth and Patrick, 2008*), we conservatively estimated that at least 700 CFU after transformation is needed to achieve 100% coverage of all 64 codon variants per position. If a transformation met the required CFU, all colonies were collected and the plasmids were purified using QIAprep Spin Miniprep Kit (QIAGEN N.V.), while those that did not were re-transformed or re-cloned until the count was met.

## Antibiotic selection of the VIM-2 mutagenized library

Mutant libraries at individual codons were mixed into seven groups of 39 (33 for the last group) consecutive codons (see 'Deep sequencing and quality control'). E. cloni 10G electrocompetent cells (Lucigen Corp.) were transformed with 1 ng of the plasmid DNA from each of the seven groups using the supplier's recommended electro-transformation protocol and grown overnight in 10 mL LB-Cm shaking at 30°C. We plated 1/1000 of the transformed culture on LB-Cm agar plates to estimate total CFU after transformation. Using CASTER and GLUE, it was estimated that 20,000 CFU are needed to fully cover 2496 codon mutants (64 codons $\times$ 39 positions) and all groups transformed had at least 100,000 CFU. The transformed libraries were suspended in LB media and preserved in 1 mL aliquots at −80°C in LB with 25% final volume glycerol.

Antibiotic selection was conducted in duplicate on two separate days. For each experiment, the 1 mL glycerol stock from each group was thawed and grown in 10 mL LB-Cm shaking at 30°C for 16 hr, with optical density at 600 nm ($OD_{600}$) of the cell culture reaching ~1.5. The cultures were then diluted by 1000 fold into fresh LB-Cm and grown at 37°C for 1.5 hr. After 1.5 hr of growth ($OD_{600}$ of the culture is between 0.01 and 0.02), 960 µL of each culture was directly introduced to 40 µL of the antibiotics at 25 $\times$ concentration (final concentrations are 128, 16 and 2.0 µg/mL for AMP, 4.0 and 0.5 µg/mL for CTX, and 0.031 µg/mL for MEM, prepared in water) or water (no selection) into the

wells of a 2.2 mL deep-well 96 well plate, and grown for 6 hr at 37°C. The cultures were also selected at the same AMP concentrations or grown without selection at 25°C. A culture of E. cloni 10G electrocompetent cells transformed with pIDR2-wtVIM-2 was also grown for 6 hr at 37°C.

After placing the cultures under selection, antibiotics and DNA from lysed cells were removed by washing the selected cultures three times using a Biomek 3000 (Beckman Coulter Inc) liquid handling robot. For each wash, the culture was centrifuged at 4000 RPM for 12 min, and the supernatant was removed using the Biomek. Subsequently, 800 µL of fresh LB was manually dispensed into the wells, the plate was sealed with plastic film, and the pellets were resuspended by vortexing. The resuspended cultures were centrifuged again for the next cycle of the wash. After the final wash, all cultures were propagated overnight shaking at 30°C in 1 mL of LB-Cm. Plasmid DNA was purified from the cultures using a QIAprep 96 Turbo Kit (QIAGEN N.V.).

### Determination of half maximal effective concentration (EC$_{50}$)

We isolated 12 variants from codon libraries at positions 55, 62, 67, 68 and 11 variants from codon libraries at positions 205, 209, 210, 211 by transforming the libraries into *E. coli,* plating on agar plates, and picking single colonies. The identity of each variant was obtained by Sanger sequencing. The variants are grown into glycerol stocks in a 2.2 mL 96 well deep well plate; two single colonies transformed with pIDR2-wtVIM-2 and empty vector were also placed on this plate as controls.

The variants in the plate were then placed under the same liquid culture selection procedure as the DMS selection experiments (see 'Antibiotic selection of the VIM-2 mutagenized library' above), up to the end of the 6 hr of selection where the cell growth ($OD_{600}$) was measured. The range of selection is 1–1024 µg/mL for AMP, 0.0625–64 µg/mL for CTX and 0.002–2 µg/mL for MEM, separated in two fold increments. All variants were also grown without antibiotics as a growth control. We calculate the $EC_{50}$ by fitting **Equation (2)** using the 'curve_fit' function of the 'Scipy.optimize' package.

$$\% \ growth = bottom + \frac{top - bottom}{1 + \left( \frac{EC50}{drug \ concentration} \right)^{\textbf{Hill Coefficient}}} \tag{2}$$

Initial estimates were 100% for top, 0% for bottom, −1.0 for the Hill coefficient and 1.0 for the $EC_{50}$. In the case where a variant's growth curve did not produce a successful fit based on the initial estimate, only the Hill coefficient and $EC_{50}$ were adjusted until the fit was successful. Variants where the $EC_{50}$ do not appear in the growth curve (stop codons, highly deleterious mutations and empty vector) could not be fitted and were excluded.

The DMS fitness scores (y-values) for each antibiotic were fitted against the $EC_{50}$ (x-values) using a similar sigmoidal curve in **Equation (3)** with the 'curve_fit' function.

$$DMS \ fitness \ score = bottom + \frac{top - bottom}{1 + \left( \frac{x_0}{EC50} \right)^{\textbf{Hill Coefficient}}} \tag{3}$$

The initial estimates for top, bottom and Hill coefficient are 2.0,–4.0 and 1.0 respectively for all antibiotics. The initial estimate for $x_0$, the inflection point of the curve, was 64 µg/mL for AMP, 4.0 µg/mL for CTX and 0.031 µg/mL for MEM.

The linear region of the sigmoidal curve for the DMS fitness scores was calculated using **Equations (4-7)**, based on the final fitted values for each antibiotic (**Sebaugh and McCray, 2003**).

$$Y_{lower} = top + \left( \frac{bottom - top}{1 + 1/4.6805} \right) \tag{4}$$

$$Y_{upper} = top + \left( \frac{bottom - top}{1 + 4.6805} \right) \tag{5}$$

$$X_{lower} = x_0 \left( \frac{bottom - Y_{lower}}{Y_{lower} - top} \right)^{\frac{1}{Hill \ Coefficient}} \tag{6}$$

$$X_{upper} = x_0 \left( \frac{bottom - Y_{upper}}{Y_{upper} - top} \right)^{\frac{1}{Hill\ Coefficient}}$$ (7)

## Deep sequencing and quality control

We grouped individual codon libraries into seven groups of 39 consecutive codons (33 for the last group) so that all mutations are within a distance of 117 bp (99 bp for the last group), allowing 150 bp forward and reverse deep sequencing reads to generate full overlap of each group. PCR amplicons of each library group and wtVIM-2 were generated using primers that flank the 117 bp region of each group, where primers have a Nextera transposase adapter sequence (Illumina, Inc) in the 5′ overhangs. We used KAPA HiFi PCR Kit (Kapa Biosystems, Inc) for 15 cycles of amplification each with denaturation at 98°C for 20 s, annealing at 65°C for 5 s and no extension time. The amplicons were extended by PCR again to include the sample indices (i7 and i5) and flow cell binding sequence, then sequenced using a NextSeq 550 sequencing system (Illumina, Inc); all samples were sequenced in the same NextSeq run. The raw sequencing data can be found on the NCBI Sequencing Read Archive (SRA) (BioProject accession: PRJNA606894). Each group under each condition received between 400,000 to 1,000,000 reads, which is at least 160 reads per codon variant on average. There were five samples that received as low as 100,000 reads in one of the two replicates, but the correlation of fitness scores between both replicates still showed an $R^2$ of at least 0.72 and the scores were retained.

To process the deep sequencing data, including merging paired-end reads, quality filtering, variant identification and fitness score calculation, we use a set of in-house Python scripts (https://github.com/johnchen93/DMS-FastQ-processing; *Chen, 2020*; copy archived at https://github.com/elifesciences-publications/DMS-FastQ-processing). Paired-end reads in FastQ format were first merged, where quality (Q) scores of matching read positions were combined using a posterior probability calculation to obtain posterior quality (Q) scores, measured on the Phred scale for sequencing quality (*Edgar and Flyvbjerg, 2015*). In the case of a base mismatch between the forward and reverse reads, the base was taken from the read with the higher Q score at the position.

Reads that had more than 20 base mismatches between the forward and reverse reads, or that had any bases with a posterior Q score less than 10 were discarded. It was found that above a posterior Q score cut-off of 10 or more, the average proportion of sequencing errors per position stabilized and no sizeable reduction of sequencing errors can be obtained by posterior Q score cut-offs (*Figure 1—figure supplement 3A*). Usually, 75–85% of all reads passed these filters. Additionally, the expected number of errors per read was calculated from adding the error rates calculated from the posterior Q Scores of every position in the entire read (*Edgar and Flyvbjerg, 2015*). Reads that had an expected number of error greater than one would also be discarded, however no reads exceeded this limit after the previous filters.

## Variant identification and noise filtering

Once forward and reverse reads were merged and filtered by read quality, codon mutants were identified and counted, then aggregated into amino acid (or stop codon) variants. Codon mutations were identified by comparing to the wtVIM-2 DNA sequence as a reference. Since we only intended for single codon mutants in the library, any sequence with mutations in more than one codon was discarded, leading to retention of 80–90% of the filtered reads.

To exclude variants that may be due to sequencing errors alone, we estimated the expected frequency of each variant generated by sequencing errors and excluded variants that have less than 2 × the expected frequency in the non-selected library. Using sequencing data from wtVIM-2, we calculated the error rates that originates from culture growth, sample preparation (PCRs) and sequencing. The error rates at each position was calculated by dividing the errors observed by the total number of reads at that position (*Figure 1—figure supplement 4D*). The mean of the distribution of the positional errors was used as the estimate for error rates (0.072%) in all positions across the VIM-2 gene. The proportion of each type of nucleotide error (A > T, A > C, A > G, etc.) was calculated to estimate the likelihood of each type of nucleotide error given a starting nucleotide (*Figure 1—figure supplement 3B*).

We made the observations that 1) sequencing errors in wt sequences will generate single codon mutants, but errors in single codon variants are most likely to be turned into double codon mutants

(*Figure 1—figure supplement 4A*) 2)~5–10% of reads in each library group were occupied by wt sequences while other variants are rarely higher than 0.5% (*Figure 1—figure supplement 4B*) and 3) single nucleotide sequencing errors are the most abundant type of errors affecting up to 10% of all reads, while higher numbers of sequencing errors are nearly negligible (*Figure 1—figure supplement 4C*). In summary, single nucleotide errors on wt sequences are the main source of single codon variants arising from sequencing errors. Thus, we first calculated the chance of each of the 64 codons to mutate into the nine adjacent codons by single nucleotide sequencing errors using *Equation (8)* (*Figure 1—figure supplement 4E*).

$$\text{chance of codon with substitution } X \rightarrow Y =$$
$$\text{error rate per position } \times \text{ frequency of substitution } X \rightarrow Y$$

(8)

For example, to gauge how often AAA gets mutated to GAA by chance, we multiplied the per position error rate by the proportion of G mutations when starting from an A, leading to 0.0719% × 70.7% = 0.000719×0.707 = 0.00051. This means for each 100,000 wt reads that has an AAA codon at a given position, we expect 51 GAA mutants on average that arise by chance at the same position. We calculated expected codon error frequencies from every codon, then summed the expected error frequencies of the codons mutant for each amino acid variant (*Figure 1—figure supplement 4F*). The error frequency is multiplied by the count of the wt reads in each non-selected library group to arrive at an expected error count for that group. Subsequently, we compared the observed count of amino acid (or codon) variants in the non-selected library to the expected count from errors alone and we accepted a variant as truly existing if the observed count is at least twice the expected count from errors. In addition, because our filtering method only accounts for the nine codons with a single mutation relative to the wt codon, we also applied a count cut-off of 5 for all variants to reduce noise by excluding very low count data.

## Fitness score calculations

The fitness score of each variant was calculated according to *Equation (1)* (see main text). To calculate the fitness score of a given amino acid (or codon) variant, the read count of the variant was first normalized to frequencies within the non-selected or selected library group. Variants that exist in the non-selected library but disappear in the selected library were interpreted to have been removed by antibiotic selection, and were given a dummy count of 1 to emulate the minimum frequency observable for that variant. The frequency of the variant after selection was divided by the frequency of the same variant in the library grown without selection for an enrichment ratio; synonymous codon variants were also considered as variants rather than wt during scoring. The variant enrichment ratio was then normalized to the enrichment ratio of the wt. The final score was expressed in $\text{Log}_2$ units, and scores were calculated separately across the seven groups and separately for each replicate.

When combining all data across the seven groups, we subtracted the mean fitness scores of all synonymous variants in each group from all variants of that group to center the mean fitness of synonymous variants at a fitness score of 0. To combine scores from replicates, we simply averaged the fitness scores across the two replicates, and take the single score if only one replicate contained the variant above noise in the non-selected library.

## Fitness effect classification

To classify each amino acid variant as positive, neutral or negative for each of the three selection antibiotics, we use the score of each variant in a two tailed z-test on a normal distribution (null-model) with the same mean and standard deviation as our synonymous distribution (244 synonymous variants total). The P-values were then FDR corrected to an $\alpha$ of 0.05 using the Benjamini-Hochberg procedure; only nonsynonymous variants were tested and the total number of tests was 5291. The variants with scores that are significantly different from the synonymous distribution after FDR correction are then classified as 'positive' if their score is greater than the synonymous mean and 'negative' if the score is less, while the remaining variants are classified as 'neutral'.

## Linear model of DMS scores with various predictors

We generated a linear model in R using a combination of terms to try and find properties that best explain the behavior we see in the DMS fitness scores. Using the fitness score as a response, we

tried using 1) wild-type (wt) amino acid 2) variant (var) amino acid 3) accessible surface area (ASA) of the residue calculated from the crystal structure of wt VIM-2 (PDB: 4bz3) using ASA view (*Ahmad et al., 2004*) 4) change in amino acid volume (*Perkins, 1986*) ($\Delta$volume = volume$_{var}$ volume$_{wt}$) 5) change in amino acid polarity (hydrophathy index *Kyte and Doolittle, 1982*) ($\Delta$polarity = $\Delta$polarity$_{var}$ – $\Delta$polarity$_{wt}$) 6) distance of the alpha carbon of each residue in the crystal structure to the active site water held between the Zn ions 7) Rosetta predicted stability change between the variant and the wt ($\Delta\Delta$G = $\Delta$G$_{var}$-$\Delta$G$_{wt}$) (also see 'Rosetta $\Delta\Delta$G Calculation' below) and 8) BLOSUM62 score for the substitution from wt to variant. Only variants from positions observable in the crystal structure were modelled (positions 32 to 262), and synonymous variants were excluded.

All parameters were first modelled individually as predictors with DMS fitness score as the response, and the predictors with $R^2$ higher than 0.10 are then modelled in combinations of two or more until the combination with the least predictors and the highest adjusted $R^2$ was found. Predictors with $R^2$ less than 0.10 are also retried in combination with the best predictors when optimizing for adjusted $R^2$. Interaction between predictors were tested, but they did not improve adjusted $R^2$ and were excluded for the sake of simplicity. The relative contribution of each term to the overall adjusted $R^2$ were calculated using the R package 'relaimpo', using the 'lmg' method (*Groemping, 2006*).

The final equation of the linear model is shown in *Equation (9)* where the fitness score of a given variant is the additive combination of the model intercept $\beta_0$ and the various properties and the coefficients of the properties (e.g. $\beta_{ASA}$ and ASA) plus a random error term $\varepsilon$.

$$\mathrm{Fitness\,Score} = \beta_0 + \beta_{\mathrm{ASA}} \times \mathrm{ASA} + \beta_{\Delta\Delta\mathrm{G}} \times \Delta\Delta\mathrm{G} + \beta_{\mathrm{wt}} \times \mathrm{wt} + \beta_{\mathrm{var}} \times \mathrm{var} + \epsilon \qquad (9)$$

The categorical predictors wt and var are simplified in the equation and each is actually a collection of terms in the model, where every amino acid is a single binary term represented by 0 or 1 such that 1 indicates the presence of the amino acid. For example, the variant Q60V has Q = 1 for wt and V = 1 for var, while all other wt and var amino acids are set to 0. Ala is not present as one of the estimates in either wt or var, because they are used in calculating the intercept; this is the mean fitness of all data points where either wt = 1 or var = 1 for alanine.

## Rosetta $\delta\delta$g calculation

To estimate the effects of each VIM-2 variant on the stability of the protein, we used the Rosetta 'ddg_monomer' application to calculate the folding energy of a monomeric protein crystal structure. Rosetta was run on the Compute Canada server Cedar using a Rosetta 3.8 installation. Following the 'ddg_monomer' documentation, the VIM-2 structure (PDB: 4bz3) was first processed using 'preminimize' to pre-optimize the packing of the crystal structure and generate a constraints file. Then, all single amino acid variant structures and the wt structure at each position were simulated 50 times each using 'ddg_monomer', configured to protocol 16 as specified in *Kellogg et al., 2011* while using Talaris 2014 as the scoring function. We store the simulated structures (variant and wt) as PDB files and scored them using the Rosetta 'score' function with Talaris 2014 weights to obtain the predicted $\Delta G$ in Rosetta Energy Units (REU). We average the predicted $\Delta G$ of all 50 replicates of each variant or wt. The $\Delta\Delta G$ is calculated using *Equation (10)* as the difference in average $\Delta G$ between variant and wt at the same position.

$$\Delta\Delta G = \Delta G_{var} - \Delta G_{wt} \qquad (10)$$

## RNA folding energy calculation for single codon mutants

The RNA folding energy contribution of the 5' UTR and signal peptide region is calculated according to a previously described method (*Bhattacharyya et al., 2018*). The $\Delta G$ of folding of the 5' UTR and signal peptide is calculated using *equation (11)*.

$$\Delta G_{1,118} = \Delta G_{1,841} - \Delta G_{119,841} \qquad (11)$$

Each $\Delta G$ term is calculated using the NUPACK software package (*Zadeh et al., 2011*), using the 'pfunc' program which calculates the $\Delta G$ of all RNA secondary structures from the partition function. All folding energies were calculated using default conditions, with [Na+]=1 M and T = 37°C, and the results are in units of kcal mol$^{-1}$. The subscripts for each $\Delta G$ term indicates the first and last

nucleotide position in the transcript that is used for calculating the $\Delta G$ of folding, respectively. Thus, $\Delta G_{1,841}$ represents the calculated folding energy of the full transcript (including the 5′ UTR and coding region, but excluding the 3′ UTR), while $\Delta G_{119,841}$ is the folding energy of the transcript after the signal peptide. The interpretation of $\Delta G_{1,118}$ is that it is the folding energy contribution of the RNA transcript up to the end of the signal peptide, including energy from non-local interactions with downstream parts of the transcript but excluding folding energy of interactions exclusively within positions downstream of the signal peptide. The DNA sequence of the wtVIM-2 transcript used to calculate the folding energies is shown below, with positions 1–118 italicized and the translated signal peptide region underlined for clarity; the sequence is converted to RNA for calculation.

5′-*CTGATAAATGCTTCAATAATATTGAAAAAGGAAGCCC<u>ATGGGATTCAAACTTTTGAGTAAG</u>* *<u>TTATTGGTCTATTTGACCGCGTCTATCATGGCTATTGCGAGCCCGCTCGCTTTTTCC</u>*GTAGATTC TAGCGGAGAATATCCGACAGTCAGCGAAATTCCGGTCGGGGGAGGTCCGGCTTTACCAGA TTGCCGATGGTGTTTGGTCGCATATCGCAACGCAGTCGTTTGATGGCGCAGTCTACCCGTCCAA TGGTCTCATTGTCCGTGATGGTGATGAGTTGCTTTTGATTGATACAGCGTGGGGTGCGAAAAACA- CAGCGGCACTTCTCGCGGAGATTGAGAAGCAAATTGGACTTCCTGTAACGCGTGCAGTC TCCACGCACTTTCATGACGACCGCGTCGGCGGCGTTGATGTCCTTCGGGCGGCTGGGG TGGCAACGTACGCATCACCGTCGACACGCCGGCTAGCCGAGGTAGAGGGGAACGAGA TTCCCACGCACTCTCTTGAAGGACTTTCATCGAGCGGGGACGCAGTGCGCTTCGGTCCAG TAGAACTCTTCTATCCTGGTGCTGCGCATTCGACCGACAACTTAATTGTGTACGTCCCGTC TGCGAGTGTGCTCTATGGTGGTTGTGCGATTTATGAGTTGTCACGCACGTCTGCGGGGAACG TGGCCGATGCCGATCTGGCTGAATGGCCCACCTCCATTGAGCGGATTCAACAACACTACCCG- GAAGCACAGTTCGTCATTCCGGGGCACGGCCTGCCGGGCGGTCTTGACTTGCTCAAGCACA- CAACGAATGTTGTAAAAGCGCACACAAATCGCTCAGTCGTTGAGTAA-3′.

*Equation (12)* is used to calculate the $\Delta\Delta G$ of folding upon codon mutations in the signal peptide.

$$\Delta\Delta G_{sigpep\,RNA} = (\Delta G_{1,118\,var} - \Delta G_{1,118\,wt})/(kT) \tag{12}$$

$\Delta G_{1,118wt}$ is the folding energy of the wtVIM-2 signal peptide sequence shown above, while $\Delta G_{1,118var}$ is the folding energy of the signal peptide sequence with a single codon mutation. The difference in folding energy is normalized to the thermal energy factor $kT$, where $k$ = 0.0019872041 kcal mol$^{-1}$ K$^{-1}$ and $T$ = 310.15 K (37°C).

## Identification of critical residues and temperature dependence

We classify the role of residues in wtVIM-2 by examining the DMS fitness scores of selection conducted at 128 µg/mL and 16 µg/mL AMP at 37°C and 25°C (four data sets), excluding signal peptide residues 1–26. Residues are classified as 'essential' when > 75% of variants are below a fitness score of −2.0 (halfway between zero and the lower score limit of −4) when selected at the least stringent condition of 16 µg/mL and 25°C. Residues are classified as 'tolerant' when > 75% of variants are above a fitness score of −1.0 (capturing the lower end of the peak centered at neutral fitness) when selected at the most stringent condition of 128 µg/mL and 37°C. Residues are classified as 'temperature dependent' when the fitness score of 25°C selection is higher than 37°C selection of the same AMP concentration by at least 2.0, for two or more variants at either 128 µg/mL or 16 µg/mL AMP. The 'temperature dependent' classifications overwrite 'essential' or 'tolerant' classifications (four occurrences total). Residues that do not fall into the three other classifications are defined as 'residue dependent'.

## Detecting hydrogen bonds in the wtVIM-2 crystal structure

Potential hydrogen bonding pairs in the wtVIM-2 crystal structure (PDB: 5yd7, chain A only) were extracted using the 'Polarpairs' script in PyMol (https://pymolwiki.org/index.php/Polarpairs). The script filters for pairs of h-bond donor and acceptor atoms within a defined distance and h-bond angle; we set the distance limit to be within 3.6 Å and the h-bond angle to be greater than 63°. The script returns atom indices, which were converted to PBD atom IDs using pymol's built in 'id_atom' function. The atoms were then extracted from the 5yd7 pdb file using the atom IDs, and the atom's name was used to determine if the h-bond was formed between backbone atoms only ('N' and 'O' atoms indicate backbone amides and carbonyls, respectively), between sidechain atoms only or

between backbone and sidechain atoms. All extracted h-bonds can be found in *Supplementary file 2G*.

## Analysis of specificity variants

We define variants with altered specificity by filtering for variants with a change in fitness effect classifications (positive, neutral and negative, defined in 'Fitness effect classification') as well as a fitness score difference of 2.0 between at least 2 of the three antibiotics being compared.

Specificity positions are visualized on the wtVIM-2 structure (PDB: 5yd7) using PyMol, with substrates overlaid from aligned MBL homolog structures (AMP - NDM-1(PDB:4hl2), cefuroxime – NDM-1(PDB:4rl0), MEM – VIM-1 (PDB:5n5i)). To overlay the substrates, the six metal binding residues (structure positions 114, 116, 118, 179, 198, 240 for VIM-1/VIM-2 and 120, 122, 124, 189, 208, 250 for NDM-1) and the two active-site Zn ions were selected from each structure, and the PyMol 'align' function was used with the VIM-2 active-site as the target object and the other structure's active-site as the mobile object. When structures have more than one chain (PDB: 5yd7, 4hl2 and 4rl0), only chain A was used in the alignment. The protein portions of the homolog structures are hidden after alignment, to visualize just the substrates with the wtVIM-2 structure.

## Collection of naturally occurring VIM variants

Amino acid sequences of naturally observed VIM variants were extracted by performing a BLASTP search of the NCBI non-redundant protein database (*NCBI Resource Coordinators, 2018*), using the protein sequence of our in-house VIM-2 with Gly removed from the 2nd position, identical to the VIM-2 discovered in *Pseudomonas aeruginosa* isolates (UniProt accession: A4GRB6). We retained all BLASTP results with at least 70% identity and >90% query coverage. We also retrieved protein sequences from the Comprehensive Antibiotic Resistance Database (CARD) (*Jia et al., 2017*). All sequences from both sources were merged and sequences that are exactly identical in length and sequence were combined, while sequences that are less than 250 or greater than 290 residues were excluded. Recombinant variants VIM-12 and VIM-25 were excluded from this analysis, as well as VIM-14 (UniProt accession: Q6GUL7) which is a member of the VIM-1 clade despite being labeled as both VIM-11 and VIM-14 (UniProt accession: A0SWU7) (both are in the VIM-2 clade).

To identify all mutations different between wtVIM-2 and the 55 other variants, a multiple sequence alignment (MSA) was constructed using the MUSCLE method in MEGA 7 (version 7.0.26) (*Kumar et al., 2016*). Equivalent positions bearing a different amino acid from VIM-2 were identified as mutations, while deletions are ignored (one in VIM-1, one in VIM-7, four in VIM-18). The MSA was also used to generate a maximum likelihood phylogenetic tree using MEGA 7 (default settings). The tree was used to identify separate VIM clades which were labeled using the VIM variant with the lowest number in the clade (VIM-1, VIM-2, VIM-7, VIM-13).

## Acknowledgements

We thank the members of the Tokuriki lab for comments on the manuscript. Canadian Institute of Health Research (CIHR) Foundation Grant to NT. NT is a Michael Smith Foundation of Health Research (MSFHR) career investigator.

## Additional information

### Funding

| Funder | Grant reference number | Author |
| --- | --- | --- |
| Canadian Institutes of Health Research | FDN-148437 | Nobuhiko Tokuriki |

The funders had no role in study design, data collection and interpretation, or the decision to submit the work for publication.

## Author contributions

John Z Chen, Conceptualization, Software, Formal analysis, Investigation, Methodology, Writing - original draft, Writing - review and editing; Douglas M Fowler, Software, Validation, Methodology, Writing - review and editing; Nobuhiko Tokuriki, Conceptualization, Supervision, Funding acquisition, Project administration, Writing - review and editing

## Author ORCIDs

John Z Chen (ID) https://orcid.org/0000-0002-2628-5820
Douglas M Fowler (ID) http://orcid.org/0000-0001-7614-1713
Nobuhiko Tokuriki (ID) https://orcid.org/0000-0002-8235-1829

## Decision letter and Author response

Decision letter https://doi.org/10.7554/eLife.56707.sa1
Author response https://doi.org/10.7554/eLife.56707.sa2

---

# Additional files

## Supplementary files

• Supplementary file 1. Plasmid map of pIDR2-wtVIM-2 in GenBank format.

• Supplementary file 2. Data and additional information. Includes the following: (A) Fitness scores for amino acid variants under all selection conditions, (B) $EC_{50}$ measured from individually isolated variants, (C) Data used in linear models, (D) Fitness scores for codon variants selected under 128 µg/mL AMP, (E) Calculated RNA folding energy of signal peptide variants, (F) Temperature dependence classifications for the catalytic domain of VIM-2, (G) Hydrogen bonds found in the 5yd7 crystal structure, (H) Variants with positive fitness (>1) in all antibiotics, (I) All identified variants with altered specificity, (J) List of all VIM-type natural variants, (K) Mutations in VIM-type natural variants relative to wtVIM-2, and (L) Primers used to extract the VIM-2 gene for deep sequencing.

• Supplementary file 3. Linear model results for variant fitness selected under CTX and MEM. (A) Variants selected under 4.0 µg/mL CTX. (B) Variants selected under 0.031 µg/mL MEM.

• Transparent reporting form

## Data availability

All processed data that are analyzed are included in the manuscript and supporting files. Raw sequencing data has been deposited in the NCBI Sequencing Read Archive and all files are submitted under the BioProject accession code PRJNA606894.

The following dataset was generated:

| Author(s) | Year | Dataset title | Dataset URL | Database and Identifier |
|---|---|---|---|---|
| Chen J, Fowler DM, Tokuriki N | 2020 | VIM-2 deep mutational scanning | https://www.ncbi.nlm.nih.gov/bioproject/?term=PRJNA606894 | NCBI BioProject, PRJNA606894 |

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
