## [Decision Letter]

**Acceptance summary:**

This comprehensive study of mutational effects on enzyme specific activity and their relationship to temperature sheds light on whether deleterious mutations exert their effect due to changes in catalytic activity or protein folding and stability. The work also addresses questions related to the impact of the signal peptide sequence on fitness and demonstrates a correlation between computed stability scores and experimental fitness values of mutations in the protein core. This very broad study addresses important questions on the relationship between sequence, structure and function in a rigorous way.

**Decision letter after peer review:**

Thank you for submitting your article "Comprehensive exploration of the translocation, stability and substrate recognition requirements in VIM-2 lactamase" for consideration by *eLife*. Your article has been reviewed by three peer reviewers, and the evaluation has been overseen by a Reviewing Editor and Michael Marletta as the Senior Editor. The following individuals involved in review of your submission have agreed to reveal their identity: Marc Ostermeier (Reviewer #1); Timothy A Whitehead (Reviewer #3).

The reviewers have discussed the reviews with one another and the Reviewing Editor has drafted this decision to help you prepare a revised submission.

We would like to draw your attention to changes in our revision policy that we have made in response to COVID-19 (https://elifesciences.org/articles/57162). Specifically, we are asking editors to accept without delay manuscripts, like yours, that they judge can stand as *eLife* papers without additional data, even if they feel that they would make the manuscript stronger. Thus the revisions requested below only address statistical analysis, clarity and presentation.

Summary:

Chen et al. sought to understand the role of each residue in the VIM-2 lactamase with respect to protein stability, activity and substrate selectivity. They generated a wealth of data by analyzing VIM-2 mutants under the selective pressures of three antibiotics across a range of concentrations. Of particular interest are the analysis of residues important for enzyme specific activity and how temperature impacts fitness. The temperature dependence analysis provides insight on whether deleterious mutations exert their effect due to changes in catalytic activity or protein folding/stability. The work also addresses questions related to the impact of the signal peptide sequence on fitness and demonstrates a correlation between computed stability scores and experimental fitness values of mutations in the protein core. The study is therefore very broad and addresses several important questions on the relationship between sequence, structure and function in a rigorous way.

The reviewers noted several areas where the statistical analysis and the presentation could be improved, as detailed below.

Essential revisions:

1) The finding that the signal peptide "is less than optimal" can arise from several reasons: the 5' UTR on the plasmid may be substantially different than that observed in clinical isolates (see recent work from Shakhnovich on how 5'UTR interactions impact fitness of coding sequences and from Ostermeier, Mehlhoff et al., 2020 on the fitness effects of TEM-1 BLA signal peptide in the absence of antibiotic selection). Hence, the authors may want to tone down the strength of their claim while suggesting alternatives or substantiate their claim through additional experiments or existing data.

2) Statistical analysis: In several points through the paper, the authors make claims on the strength of selection regarding specific mutations. These claims should be substantiated by a statistical measure because they may result from low sequence coverage in the unselected populations as is often the case in large-scale mutational screens such as this. In addition, the lack of uncertainty values for each fitness score limits the usefulness of this data set for other researchers, should they want to use values from individual mutations instead of looking at collective or global effects of mutations. Examples:

a) Subsection “Codon and amino acid optimization in the signal peptide”, second paragraph: The authors note that only mutations to Cys or Trp are deleterious at Leu23. Are the fitness values of Leu23Cys and Leu23Trp statistically different than neutral?

b) Subsection “Codon and amino acid optimization in the signal peptide”, last paragraph: The first sentence talks about Figure 5C and how the WT residues or conserved residues do not have the highest fitness. But the authors provide no statistical test to say whether these fitness scores are different than each other, and many of them fall within the range of -0.7 to 0.7 that are considered neutral. This same comment applies throughout this paragraph. Are differences in values that the authors cite supported by statistical tests? Such statistical analysis is necessary for the claims made. This is particularly true when comparing synonymous mutations because the sequencing counts are lower than when counts of all synonymous mutations are grouped.

c) Subsection “Distinct recognition for different classes of β-lactam substrates”, first paragraph: Statements on purported adaptive mutants at positions 47, etc. should be substantiated by a statistical test using the uncertainty in those fitness values or by performing *EC*_50_ assays, as was done to substantiate claims of changes in substrate specificity.

d) Supplementary Data: The authors provide the fitness scores but no measure of the uncertainty in those scores. Such an analysis of the uncertainty is necessary for several claims made throughout the paper.

e) The comparison to *EC*_50_ between deep sequencing results and individual clones should be supported by some statistical measure, possibly a nonparametric rank-order correlation. Additionally, their "*EC*_50_" should more likely be classified as an *IC*_50_, and the half-inhibitory concentration they are measuring is probably not very accurate.

3) Given that there is now a better understanding for allowed mutations within Vim2 and a fitness description, would the authors be able to suggest measures about treatment options? For instance co-administration or alteration of different antibiotics?

---

## [Author Response]

Essential revisions:1) The finding that the signal peptide "is less than optimal" can arise from several reasons: the 5' UTR on the plasmid may be substantially different than that observed in clinical isolates (see recent work from Shakhnovich on how 5'UTR interactions impact fitness of coding sequences and from Ostermeier, Mehlhoff et al., 2020 on the fitness effects of TEM-1 BLA signal peptide in the absence of antibiotic selection). Hence, the authors may want to tone down the strength of their claim while suggesting alternatives or substantiate their claim through additional experiments or existing data.

Indeed, we expect the 5’UTR to be a factor in the codon level fitness and we believe we have cited the work from Shakhnovich (PMID: 29883608); the RNA folding energy calculation in our study follows their methodology. We have softened the language in our claim and explicitly made mention to this caveat in the concluding paragraph (subsection “Codon and amino acid optimization in the signal peptide”) of this section to deliver a more accurate message, in which we now discuss the sensitivity of the signal peptide to change, instead of declaring an absolute lack of optimization.

The work from the Ostermeier group has also been integrated into the subsection “Global view of VIM-2 enzyme characteristics” (deleterious effects of Cys mutations) and in the second and last paragraphs of the subsection “Codon and amino acid optimization in the signal peptide” (disruption of the hydrophobic region by charged residues) and (effects on translocation).

2) Statistical analysis: In several points through the paper, the authors make claims on the strength of selection regarding specific mutations. These claims should be substantiated by a statistical measure because they may result from low sequence coverage in the unselected populations as is often the case in large-scale mutational screens such as this. In addition, the lack of uncertainty values for each fitness score limits the usefulness of this data set for other researchers, should they want to use values from individual mutations instead of looking at collective or global effects of mutations. Examples:

Comments 2a-c address a common theme in the lack of specific support for differences in fitness scores. In response, we have made more explicit reference to the classification of positive/neutral/negative fitness effects relative to wtVIM-2, which is supported by a z-test against the synonymous variants at 5% FDR correction using the Benjamini-Hochberg procedure. This classification scheme will be used whenever we wish to evaluate the effect of variants relative to wtVIM-2.

In addition, we have included extra statistical support for claims based on differences between distributions using two-tailed Mann-Whitney U tests:

i) The general inference of Cys and Pro variants displaying more deleterious effects than mutations to other amino acids, in the first paragraph of the subsection “Global view of VIM-2 enzyme characteristics”, is now supported by Mann-Whitney U tests between all unique pairs of missense variant distributions grouped by mutated amino acid (Figure 3—figure supplement 3).

ii) The claim that buried positions are more sensitive to mutation, in the aforementioned paragraph, has been supported by a test between the DFE of variants with ASA <0.3 and the DFE of variants with ASA ≥ 0.3. The distributions and test results have also been added to Figure 4A.

iii) The claim that the signal peptide is tolerant to mutations, in the second paragraph of the subsection “Codon and amino acid optimization in the signal peptide”, is now supported by a test between the DFEs of the signal peptide and the body of VIM-2. The discussion of tolerated signal peptide mutations has also been limited to missense variants to exclude stop codon variants and the related information has been updated in Figure 5B and in the aforementioned paragraph.

a) Subsection “Codon and amino acid optimization in the signal peptide”, second paragraph: The authors note that only mutations to Cys or Trp are deleterious at Leu23. Are the fitness values of Leu23Cys and Leu23Trp statistically different than neutral?

The mentioned mutations are the strongest of 5 negative variants out of 95 neutral or positive variants. We have noted the above in the second paragraph of the subsection “Codon and amino acid optimization in the signal peptide” and have provided the fitness scores of the L23C and L23W variants as support.

b) Subsection “Codon and amino acid optimization in the signal peptide”, last paragraph: The first sentence talks about Figure 5C and how the WT residues or conserved residues do not have the highest fitness. But the authors provide no statistical test to say whether these fitness scores are different than each other, and many of them fall within the range of -0.7 to 0.7 that are considered neutral. This same comment applies throughout this paragraph. Are differences in values that the authors cite supported by statistical tests? Such statistical analysis is necessary for the claims made. This is particularly true when comparing synonymous mutations because the sequencing counts are lower than when counts of all synonymous mutations are grouped.

The exact proportion of variants in the signal peptide with a positive effect relative to wtVIM-2 (10%) and the range of scores observed for these variants (0.7-1.3) have been noted in the third paragraph of the subsection “Codon and amino acid optimization in the signal peptide”.

We have removed the claim for increased and decreased fitness in individual codon variants, as we do not have enough power to conduct a statistical test using just two replicates.

We agree that the sentence describing the ‘codon dependent variants’ could be subject to improvement by a statistical test between synonymous codon mutants rather than a score difference cut-off. However, we only aim to note a trend where the majority of codon variation lies in the signal peptide and this trend would most likely remain similar even with a more stringent statistical test. We have weakened our statement to match the lack of statistical testing.

c) Subsection “Distinct recognition for different classes of β-lactam substrates”, first paragraph: Statements on purported adaptive mutants at positions 47, etc. should be substantiated by a statistical test using the uncertainty in those fitness values or by performing EC_50_ assays, as was done to substantiate claims of changes in substrate specificity.

As described above, variants with positive effects were supported by a statistical test against the synonymous distribution as variants with fitness scores > 0.7. In this claim, we took a more conservative cut-off of fitness score > 1 in all antibiotics, which was chosen to be above the upper fitness score range of the peak centered at neutral fitness in the DFE (Figure 2, center column); this rationale has been clarified in the first paragraph of the subsection “Distinct recognition for different classes of β-lactam substrates”. Furthermore, the strong correlation between fitness and *EC*_50_ of 39-45 variants in Figure 2 (right column) provides confidence that variants with high fitness scores also have increased resistance.

With regards to the term ‘adaptive’, we have further clarified that these mutants are those with positive fitness effects in the aforementioned paragraph. The term ‘adaptive’ has been modified to ‘positive’ to reflect the observed behavior in our dataset and avoid suggestions of evolutionary adaptation, which we agree would require further testing.

d) Supplementary Data: The authors provide the fitness scores but no measure of the uncertainty in those scores. Such an analysis of the uncertainty is necessary for several claims made throughout the paper.

As suggested, we included the sample standard deviation for both amino acid and codon variant fitness scores in Supplementary file 2A and D). As noted above, since most of our analysis involve relative differences to wtVIM-2, we’ve made an effort to adhere to the statistical test for positive (fitness >0.7), neutral (fitness from -0.7 to 0.7) or negative (fitness <-0.7) fitness effects relative to wtVIM-2 in those cases. For case where the classifications do not apply, such as the comparison of mutations at the codon level, we have made other adjustments in response to those comments.

e) The comparison to EC_50_ between deep sequencing results and individual clones should be supported by some statistical measure, possibly a nonparametric rank-order correlation. Additionally, their "EC_50_" should more likely be classified as an IC_50_, and the half-inhibitory concentration they are measuring is probably not very accurate.

As suggested, the comparisons between *EC*_50_ and fitness have been supported using a Spearman rank-order correlation. To accommodate a direct relation of rank between fitness and *EC*_50_ for each variant, Figure 2 has also been modified – where the correlation plots previously compared fitness to all uniquely measured *EC*_50_ values, fitness is now correlated to the average *EC*_50_ of each unique amino acid variant. This leads to some minor changes in the optimized sigmoidal fit parameters, which have been updated in the last paragraph of the subsection “Deep mutational scanning of VIM-2 metallo-β-lactamase”.

We prefer the use of *EC*_50_ over *IC*_50_ due to the bactericidal nature of the β-lactam antibiotics. Even though the mechanism of the β-lactam is inhibition of Penicillin-binding proteins (PBP), the response we are measuring to infer resistance is not an inhibition of growth but rather death due to PBP inhibition. As such, we believe the more general term of *EC*_50_ is more appropriate.

3) Given that there is now a better understanding for allowed mutations within Vim2 and a fitness description, would the authors be able to suggest measures about treatment options? For instance co-administration or alteration of different antibiotics?

We have extended the discussion in the last paragraph of the subsection “Conclusion” to include a suggestion for further investigation of combined usage of different classes of antibiotics. Ultimately, the situation involving multi-drug resistance is complex and MBLs such as VIM-2 are only a part of the entire problem. Hence, we also note various caveats and other solutions that need to be considered.